# SimMTM: A Simple Pre-Training Framework for Masked Time-Series Modeling

**Jiaxiang Dong,**\* **Haixu Wu,**\* **Haoran Zhang, Li Zhang, Jianmin Wang, Mingsheng Long**[✉]
School of Software, BNRist, Tsinghua University, China
`{djx20,z-hr20}@mails.tsinghua.edu.cn, wuhaixu98@gmail.com`
`{lizhang,jimwang,mingsheng}@tsinghua.edu.cn`

## Abstract

Time series analysis is widely used in extensive areas. Recently, to reduce labeling expenses and benefit various tasks, self-supervised pre-training has attracted immense interest. One mainstream paradigm is masked modeling, which successfully pre-trains deep models by learning to reconstruct the masked content based on the unmasked part. However, since the semantic information of time series is mainly contained in temporal variations, the standard way of randomly masking a portion of time points will seriously ruin vital temporal variations of time series, making the reconstruction task too difficult to guide representation learning. We thus present SimMTM, a Simple pre-training framework for Masked Time-series Modeling. By relating masked modeling to manifold learning, SimMTM proposes to recover masked time points by the weighted aggregation of multiple neighbors outside the manifold, which eases the reconstruction task by assembling ruined but complementary temporal variations from multiple masked series. SimMTM further learns to uncover the local structure of the manifold, which is helpful for masked modeling. Experimentally, SimMTM achieves state-of-the-art fine-tuning performance compared to the most advanced time series pre-training methods in two canonical time series analysis tasks: forecasting and classification, covering both in- and cross-domain settings. Code is available at https://github.com/thuml/SimMTM.

## 1 Introduction

Time series analysis has attached immense importance in extensive real-world applications, such as financial analysis, energy planning and etc [47, 51]. Vast amounts of time series are incrementally collected from IoT and wearable devices. However, the semantic information of time series is mainly buried in human-indiscernible temporal variations, making it difficult to annotate. Recently, self-supervised pre-training has been widely explored [23, 15], which benefits deep models from pretext knowledge learned over large-scale unlabeled data and further promotes the performance of various downstream tasks. Mainly, as a well-recognized pre-training paradigm, masked modeling has achieved great success in many areas, such as masked language modeling (MLM) [7, 31, 32, 3, 10] and masked image modeling (MIM) [11, 50, 20]. This paper extends pre-training methods to time series, especially masked time-series modeling (MTM).

The canonical technique of masked modeling is to optimize the model by learning to reconstruct the masked content based on the unmasked part [7]. However, unlike images and natural languages whose patches or words contain much even redundant semantic information, we observe that the valuable semantic information of time series is mainly included in the temporal variations, such as the trend, periodicity, and peak valley, which can correspond to unique weather processes, abnormal faults, etc. in the real world. Therefore, directly masking a portion of time points will seriously ruin

---

\*Equal Contribution

37th Conference on Neural Information Processing Systems (NeurIPS 2023).

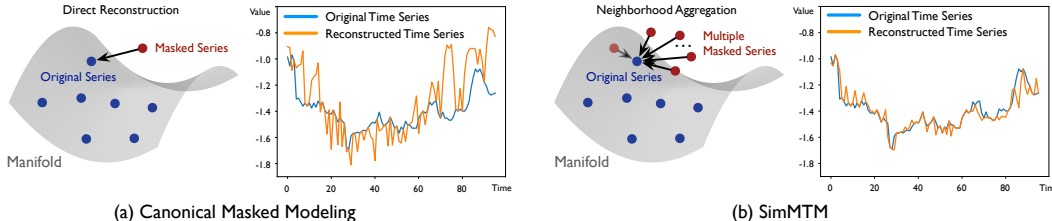

Figure 1: Comparison between (a) canonical masked modeling and (b) SimMTM in both manifold perspective and reconstruction performance. The showcase is to recover 50% masked time series.

the temporal variations of the original time series, which makes the reconstruction task too difficult to guide representation learning of time series. (Figure 1).

According to the analysis in stacked denoising autoencoders [40], as shown in Figure 1, we can view the randomly masked series as the "neighbor" of the original time series outside the manifold and the reconstruction process is to project the masked series back to the manifold of original series. However, as we analyzed above, direct reconstruction may fail since the essential temporal variations are ruined by random masking. Inspired by the manifold perspective, we go beyond the straightforward reconstruction convention of masked modeling and propose a natural idea of reconstructing the original data from its *multiple* "neighbors", i.e. multiple masked series. Although the temporal variations of the original time series have been partially dropped in each randomly masked series, the multiple randomly masked series will complement each other, making the reconstruction process much more accessible than directly reconstructing the original series from a single masked series. This process will also pre-train the model to uncover the local structure of the time series manifold implicitly, thereby benefiting masked modeling and representation learning [35, 41].

Based on the above insights, we propose the SimMTM as a simple but effective pre-training framework for time series in this paper. Instead of directly reconstructing the masked time points from unmasked parts, SimMTM recovers the original time series from multiple randomly masked time series. Technically, SimMTM presents a neighborhood aggregation design for reconstruction, which is to aggregate the point-wise representations of time series based on similarities learned in the series-wise representation space. In addition to the reconstruction loss, a constraint loss is presented to guide the series-wise representation learning based on the neighborhood assumption of the time series manifold. Empowering by above designs, SimMTM achieves consistent state-of-the-art in various time series analysis tasks when fine-tuning the pre-trained model into downstream tasks, covering both the low-level forecasting and high-level classification tasks, even if there is a clear domain shift between pre-training and fine-tuning datasets. Overall, our contributions are summarized as follows:

- Inspired by the manifold perspective of masking, we propose a new task for masked time-series modeling, which is to reconstruct the original time series on the manifold based on multiple masked series outside the manifold.
- Technically, we present SimMTM as a simple but effective pre-training framework, which aggregates point-wise representations for reconstruction based on the similarities learned in series-wise representation space.
- SimMTM consistently achieves the state-of-the-art fine-tuning performance in typical time series analysis tasks, including low-level forecasting and high-level classification, covering both in- and cross-domain settings.

## 2 Related Work

### 2.1 Self-supervised Pre-training

Self-supervised pre-training is an important research topic for learning generalizable and shared knowledge from large-scale data and further benefiting downstream tasks [15]. Originally, this topic has been widely explored in computer vision and natural language processing. Elaborative manually-designed self-supervised tasks are presented, which can be roughly categorized into contrastive learning [12, 5, 4] and masked modeling [7, 11]. Recently, following the well-established contrastive learning and masked modeling paradigms, some self-supervised pre-training methods for time series have been proposed [9, 27, 34, 33, 36, 54, 52, 6].

**Contrastive learning.** The critical insight of contrastive learning is to optimize the representation space based on the manually designed positive and negative pairs. where representations of positive pairs are optimized to be close to each other. In contrast, negative ones tend to be far apart [48, 14]. The canonical design presented in SimCLR [37] views different augmentations of the same sample as positive pairs and augmentations among different samples as negative pairs.

Recently, in time series pre-training, many designs of positive and negative pairs have been proposed by utilizing the invariant properties of time series. Concretely, to make the representation learning seamlessly related to temporal variations, TimCLR [53] adopts the DTW [25] to generate phase-shift and amplitude-change augmentations, which is more suitable for time series context. TS2Vec [55] splits multiple time series into several patches and further defines the contrastive loss in both instance-wise and patch-wise aspects. TS-TCC [8] presents a new temporal contrastive learning task as making the augmentations predict each other's future. Mixing-up [45] exploits a data augmentation scheme in which new samples are generated by mixing two data samples. LaST [42] aims to disentangle the seasonal-trend representations in the latent space based on variational inference. Afterward, CoST [46] employs contrastive losses in both time and frequency domain to learn discriminative seasonal and trend representations. Besides, TF-C [57] proposes a novel time-frequency consistency architecture and optimizes time-based and frequency-based representations of the same example to be close to each other. Note that contrastive learning mainly focuses on the high-level information [49], and the series-wise or patch-wise representations inherently mismatch the low-level tasks, such as time series forecasting. Thus, in this paper, we focus on the masked modeling paradigm.

**Masked modeling.** The masked modeling paradigm optimizes the model by learning to reconstruct the masked content from the unmasked part. This paradigm has been widely explored in computer vision and natural language processing, which is to predict the masked words of a sentence [7] and masked patches of an image [2, 11, 50] respectively. As for the time series analysis, TST [56] follows the canonical masked modeling paradigm and learns to predict removed time points based on the remaining time points. Next, PatchTST [26] proposes to predict masked subseries-level patches to capture the local semantic information and reduce memory usage. Ti-MAE [21] uses mask modeling as an auxiliary task to boost the forecasting and classification performances of advanced Transformer-based methods. However, as we stated before, directly masking time series will ruin the essential temporal variations, making the reconstruction too difficult to guide the representation learning. Unlike the direct reconstruction in previous works, SimMTM presents a new masked modeling task, which is reconstructing the original time series from multiple randomly masked series.

## 2.2 Understanding Masked Modeling

Masked modeling has been explored in stacked denoising autoencoders [40], where masking is viewed as adding noise to the original data and the masked modeling is to project masked data from the neighborhood back to the original manifold, namely denoising. It has recently been widely used in pre-training, which can learn valuable low-level information from data unsupervisedly [49]. Inspired by the manifold perspective, we go beyond the classical denoising process and project the masked data back to the manifold by aggregating multiple masked time series within the neighborhood.

## 3 SimMTM

As aforementioned, to tackle the problem that randomly masking time series will ruin the essential temporal variation information, SimMTM proposes to reconstruct the original time series from multiple masked time series. To implement this, SimMTM first learns similarities among multiple time series in the series-wise representation space and then aggregates the point-wise representations of these time series based on pre-learned series-wise similarities. Next, we will detail the techniques in both model architecture and pre-training protocol aspects.

### 3.1 Overall Architecture

The reconstruction process of SimMTM involves the following four modules: masking, representation learning, series-wise similarity learning and point-wise reconstruction.

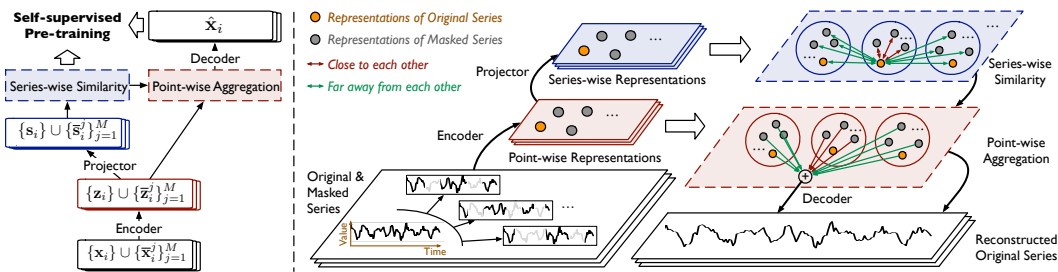

Figure 2: Architecture of SimMTM, which reconstructs the original time series by adaptive aggregating multiple masked time series based on series-wise similarities learned contrastively from data.

**Masking.** Given $\{\mathbf{x}_i\}_{i=1}^N$ as a mini-batch of $N$ time series samples, where $\mathbf{x}_i \in \mathbb{R}^{L \times C}$ contains $L$ time points and $C$ observed variates, we can easily generate a set of masked series for each sample $\mathbf{x}_i$ by randomly masking a portion of time points along the temporal dimension, formalizing by:

$$\{\overline{\mathbf{x}}_i^j\}_{j=1}^M = \mathrm{Mask}_r(\mathbf{x}_i), \tag{1}$$

where $r \in [0, 1]$ denotes the masked portion. $M$ is a hyperparameter for the number of masked time series. $\overline{\mathbf{x}}_i^j \in \mathbb{R}^{L \times C}$ represents the $j$-th masked time series of $\mathbf{x}_i$, where the values of masked time points are replaced by zeros. With above process, we can obtain a batch of augmented time series. For clarity, we present all the $(N(M+1))$ input series in a set as $\mathcal{X} = \bigcup_{i=1}^N \left(\{\mathbf{x}_i\} \cup \{\overline{\mathbf{x}}_i^j\}_{j=1}^M\right)$.

**Representation learning.** After the encoder and projector layer, we can obtain the point-wise representations $\mathcal{Z}$ and series-wise representations $\mathcal{S}$, which can be formalized as:

$$\mathcal{Z} = \bigcup_{i=1}^N \left(\{\mathbf{z}_i\} \cup \{\overline{\mathbf{z}}_i^j\}_{j=1}^M\right) = \mathrm{Enocder}(\mathcal{X}), \quad \mathcal{S} = \bigcup_{i=1}^N \left(\{\mathbf{s}_i\} \cup \{\overline{\mathbf{s}}_i^j\}_{j=1}^M\right) = \mathrm{Projector}(\mathcal{Z}), \quad (2)$$

where $\mathbf{z}_i, \overline{\mathbf{z}}_i^j \in \mathbb{R}^{L \times d_{\mathrm{model}}}$ and $\mathbf{s}_i, \overline{\mathbf{s}}_i^j \in \mathbb{R}^{1 \times d_{\mathrm{model}}}$. $\mathrm{Encoder}(\cdot)$ denotes the model encoder, which can project the input data into deep representations and will be transferred to downstream tasks during the fine-tuning process. In this paper, we implement the encoder as a well-acknowledged Transformer [39] and ResNet [13]. As for the $\mathrm{Projector}(\cdot)$, we employ a simple MLP layer along the temporal dimension to obtain series-wise representations. More details can be found in Section 4. Technically, the encoder is applied to input series separately, namely $\bigcup_{i=1}^N (\mathrm{Encoder}(\mathbf{x}_i) \cup \{\mathrm{Encoder}(\overline{\mathbf{x}}_i^j)\}_{j=1}^M)$, and so does for projector. Here we adopt the set-style formalization for conciseness.

**Series-wise similarity learning.** Note that directly averaging multiple masked time series will result in the over-smoothing problem [40], impeding the representation learning. Thus, to precisely reconstruct the original time series, we attempt to utilize the similarities among series-wise representations $\mathcal{S}$ for weighted aggregation, namely exploiting the local structure of the time series manifold. For simplification, we formalize the calculation of series-wise similarities as follows:

$$\mathbf{R} = \mathrm{Sim}(\mathcal{S}) \in \mathbb{R}^{D \times D}, D = N(M+1), \quad \mathbf{R}_{\mathbf{u},\mathbf{v}} = \frac{\mathbf{u}\mathbf{v}^\top}{\|\mathbf{u}\|\|\mathbf{v}\|}, \mathbf{u}, \mathbf{v} \in \mathcal{S}, \tag{3}$$

where $\mathbf{R}$ is the matrix of pair-wise similarities for $(N(M+1))$ input samples in series-wise representation space, which are measured by the cosine distance. $\mathbf{R}_{\mathbf{u},\mathbf{v}}$ is the calculated similarity between series-wise representations $\mathbf{u}, \mathbf{v} \in \mathcal{S}$.

**Point-wise aggregation.** As shown in Figure 2, based on the learned series-wise similarities, the aggregation process for the $i$-th original time series is:

$$\widehat{\mathbf{z}}_i = \sum_{\mathbf{s}' \in \mathcal{S} \backslash \{\mathbf{s}_i\}} \frac{\exp(\mathbf{R}_{\mathbf{s}_i,\mathbf{s}'}/\tau)}{\sum_{\mathbf{s}'' \in \mathcal{S} \backslash \{\mathbf{s}_i\}} \exp(\mathbf{R}_{\mathbf{s}_i,\mathbf{s}''}/\tau)} \mathbf{z}', \tag{4}$$

where $\mathbf{z}'$ represents the corresponding point-wise representation of $\mathbf{s}'$, i.e. $\mathbf{z}' = \mathrm{Projector}(\mathbf{s}')$. $\widehat{\mathbf{z}}_i \in \mathbb{R}^{L \times d_{\mathrm{model}}}$ is the reconstructed point-wise representation. $\tau$ denotes the temperature hyperparameter

of softmax normalization for series-wise similarities. It is notable that as described in Eq. (4), for each time series $\mathbf{x}_i$, the reconstruction is not only based its own masked series $\{\overline{\mathbf{x}}_i^j\}_{j=1}^M$. We also introduce other series representations $\mathcal{S}\backslash\{\mathbf{s}_i\}$ into aggregation, which requires the model to suppress the interference of less-related noise series and precisely learn similar representations for both the masked and the original series, namely guiding the model to learn the manifold structure better. After the decoder, we can obtain the reconstructed original time series, namely

$$\{\widehat{\mathbf{x}}_i\}_{i=1}^N = \text{Decoder}(\{\widehat{\mathbf{z}}_i\}_{i=1}^N), \tag{5}$$

where $\widehat{\mathbf{x}}_i \in \mathbb{R}^{L \times C}$ is the reconstruction to $\mathbf{x}_i$. $\text{Decoder}(\cdot)$ is instantiated as a simple MLP layer along the channel dimension following [50].

### 3.2 Self-supervised Pre-training

Following the masked modeling paradigm, SimMTM is supervised by a reconstruction loss:

$$\mathcal{L}_{\text{reconstruction}} = \sum_{i=1}^N \|\mathbf{x}_i - \widehat{\mathbf{x}}_i\|_2^2. \tag{6}$$

Note that the reconstruction process is directly based on the series-wise similarities, while it is hard to guarantee the model captures the precise similarities without explicit constraints in the series-wise representation space. Thus, to avoid trivial aggregation, we also utilize the neighborhood assumption of the time series manifold to calibrate the structure of series-wise representation space $\mathcal{S}$. For clarity, we formalize the neighborhood assumption by defining positive and negative pairs as follows:

$$\begin{aligned} \text{Positive pairs: } &(\mathbf{s}_i, \mathbf{s}_i^+), \ \mathbf{s}_i^+ \in \{\overline{\mathbf{s}}_i^j\}_{j=1}^M \\ \text{Negative pairs: } &(\mathbf{s}_i, \mathbf{s}_i^-), \ \mathbf{s}_i^- \in \{\mathbf{s}_k\} \cup \{\overline{\mathbf{s}}_k^j\}_{j=1}^M, i \neq k \end{aligned} \tag{7}$$

where $\mathbf{s}_i^+$ and $\mathbf{s}_i^-$ mean the elements that are assumed as close to and far away from $\mathbf{s}_i$ respectively. Eq. (7) indicates that the original time series and its masked series will present close representations and be far away from the representations from other series in $\mathcal{S}$. For each series-wise representation $\mathbf{s} \in \mathcal{S}$, we define the set of its assumed close series as $\mathcal{S}^+ \subset \mathcal{S}$. Note that to avoid the dominating representation, we assume that $\mathbf{s} \notin \mathcal{S}^+$. With the above formalization, we can define manifold constraint to series-wise representation space as:

$$\mathcal{L}_{\text{constraint}} = -\sum_{\mathbf{s}\in\mathcal{S}} \left( \sum_{\mathbf{s}'\in\mathcal{S}^+} \log \frac{\exp(\mathbf{R}_{\mathbf{s},\mathbf{s}'}/\tau)}{\sum_{\mathbf{s}''\in\mathcal{S}\backslash\{\mathbf{s}\}} \exp(\mathbf{R}_{\mathbf{s},\mathbf{s}''}/\tau)} \right), \tag{8}$$

which can optimize the learned series-wise representations to satisfy the neighborhood assumption in Eq. (7) better. Finally, the overall optimization process of SimMTM can be represented as follows:

$$\min_{\Theta} \mathcal{L}_{\text{reconstruction}} + \lambda \mathcal{L}_{\text{constraint}}, \tag{9}$$

where $\Theta$ denotes the set of all parameters of the deep architecture. To trade off the two parts in Eq. (9), we adopt the tuning strategy presented by [17], which can adjust the hyperparameters $\lambda$ adaptively according to the homoscedastic uncertainty of each loss.

## 4 Experiments

To fully evaluate SimMTM, we conduct experiments on two typical time series analysis tasks: forecasting and classification, covering low-level and high-level representation learning. Further, we present the fine-tuning performance for each task under in- and cross-domain settings.

**Benchmarks.** We summarize the experiment benchmarks in Table 1, comprising twelve real-world datasets that cover two mainstream time series analysis tasks: time series forecasting and classification. Concretely, we have followed the standard experimental setups in Autoformer [47] for the forecasting tasks and the experiment settings proposed by TF-C [57] for classification.

**Implementations.** We compare SimMTM with six competitive state-of-the-art baseline methods, including the contrastive learning methods: TF-C [57], CoST [46], TS2Vec [55], LaST [42], the masked modeling method: Ti-MAE [21], TST [56]. TF-C [57] and Ti-MAE [21] are the previous best pre-training methods. We experiment on both in- and cross-domain settings. For the in-domain setting, we pre-train and fine-tune the model using the same dataset. As for the cross-domain setting, we pre-train the model on a certain dataset and fine-tune the encoder to different datasets. More details are provided in Appendix A.

Table 1: Summary of experiment benchmarks.

| Tasks | Datasets | Semantic |
|---|---|---|
| Forecasting | ETTh1,ETTh2 | Electricity |
| | ETTm1,ETTm2 | Electricity |
| | Weather | Weather |
| | Electricity | Electricity |
| | Traffic | Transportation |
| Classification | SleepEEG | EEG |
| | Epilepsy | EEG |
| | FD-B | Faulty Detection |
| | Gesture | Hand Movement |
| | EMG | Muscle Responses |

**Unified encoder.** To make a fair comparison, we attempt to unify encoder for these pre-training methods. Concretely, we adopt vanilla Transformer [39] with the channel independence [26] as the unified encoder for forecasting. The channel-independent design allows models to accomplish cross-domain transfer between datasets with different variate numbers. As for the classification, we use 1D-ResNet [13] as the shared encoder following [57]. Notably, for LaST [42] and TFC [57], since their designs are closely related to model structures, we directly report results from their papers or reproduce with official codes. For other baselines and SimMTM, the results in the main text are from the unified encoder. The results from their official papers are also compared in Appendix A.2. For all baselines, the results with a unified encoder generally surpass results reported by themselves.

## 4.1 Main results

We summarize the results in forecasting and classification of in- and cross-domain settings in Figure 3. In all these settings, SimMTM outperforms other baselines significantly. It is also notable that although the masking-based method Ti-MAE [21] achieves good performance in the forecasting task (x-axis of Figure 3), it fails in the classification task (y-axis). Besides, contrastive-based methods fail in low-level forecasting tasks. These results indicate that previous methods cannot simultaneously cover high-level and low-level tasks, highlighting the advantages of SimMTM in task generality.

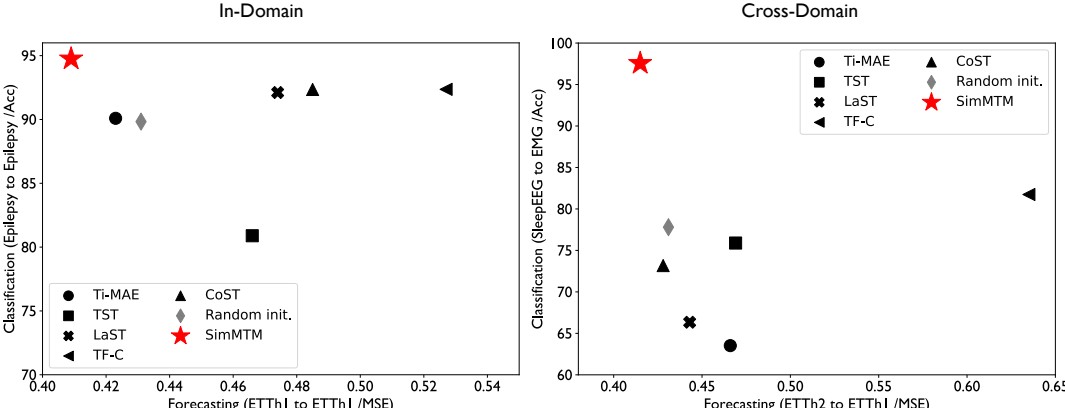

Figure 3: Performance comparison of time series pre-training methods in forecasting (*MSE* ↓) and classification (*Acc* ↑) tasks, including both in-domain (left) and cross-domain (right) settings.

## 4.2 Forecasting

**In-domain.** As shown in Table 2, empowering by SimMTM pre-training, the model performance is promoted significantly (SimMTM vs. Random init.). Besides, SimMTM consistently outperforms other pre-training methods. On the average of all benchmarks, SimMTM achieves 8.3% MSE reduction and 4.3% MAE reduction compared to the advanced masked modeling baseline Ti-MAE [21],

Table 2: In-domain setting of forecasting the future $O$ time points based on the past 336 time points. All results are averaged from 4 different choices of $O \in \{96, 192, 336, 720\}$. A smaller MSE or MAE indicates a better prediction. See Appendix E for full results.

| Models | SimMTM | | Random init. | | Ti-MAE [21] | | TST [56] | | LaST [42] | | TF-C [57] | | CoST [46] | | TS2Vec [55] | |
|---|---|---|---|---|---|---|---|---|---|---|---|---|---|---|---|---|
| Metric | MSE | MAE | MSE | MAE | MSE | MAE | MSE | MAE | MSE | MAE | MSE | MAE | MSE | MAE | MSE | MAE |
| ETTh1 | **0.409** | **0.428** | 0.431 | 0.448 | 0.423 | 0.446 | 0.466 | 0.462 | 0.474 | 0.461 | 0.527 | 0.513 | 0.485 | 0.472 | 0.446 | 0.456 |
| ETTh2 | **0.353** | **0.390** | 0.395 | 0.427 | 0.380 | 0.386 | 0.404 | 0.421 | 0.449 | 0.459 | 0.692 | 0.724 | 0.399 | 0.427 | 0.417 | 0.468 |
| ETTm1 | **0.348** | **0.385** | 0.356 | 0.387 | 0.366 | 0.391 | 0.373 | 0.389 | 0.398 | 0.398 | 0.496 | 0.474 | 0.356 | 0.385 | 0.699 | 0.557 |
| ETTm2 | **0.263** | **0.320** | 0.279 | 0.336 | 0.267 | 0.325 | 0.297 | 0.347 | 0.265 | 0.327 | 0.465 | 0.562 | 0.314 | 0.365 | 0.326 | 0.361 |
| Weather | **0.230** | 0.271 | 0.239 | 0.275 | 0.234 | **0.265** | 0.239 | 0.276 | 0.232 | 0.261 | 0.286 | 0.349 | 0.324 | 0.329 | 0.233 | 0.267 |
| Electricity | **0.162** | **0.256** | 0.212 | 0.300 | 0.205 | 0.296 | 0.209 | 0.289 | 0.186 | 0.274 | 0.363 | 0.432 | 0.215 | 0.295 | 0.213 | 0.293 |
| Traffic | **0.392** | **0.264** | 0.490 | 0.316 | 0.475 | 0.310 | 0.586 | 0.362 | 0.713 | 0.397 | 0.717 | 0.456 | 0.435 | 0.362 | 0.470 | 0.350 |
| Avg | **0.308** | **0.331** | 0.343 | 0.356 | 0.336 | 0.346 | 0.368 | 0.364 | 0.388 | 0.368 | 0.507 | 0.501 | 0.361 | 0.376 | 0.401 | 0.393 |

Table 3: Cross-domain setting of forecasting the future $O$ time points based on the past 336 time points. All results are averaged from 4 different choices of $O \in \{96, 192, 336, 720\}$. A lower MSE or MAE indicates a better prediction. Full results are in Appendix E.

| Models | SimMTM | | Ti-MAE [21] | | TST [56] | | LaST [42] | | TF-C [57] | | CoST [46] | | TS2Vec [55] | |
|---|---|---|---|---|---|---|---|---|---|---|---|---|---|---|
| Metric | MSE | MAE | MSE | MAE | MSE | MAE | MSE | MAE | MSE | MAE | MSE | MAE | MSE | MAE |
| ETTh2 → ETTh1 | **0.415** | **0.430** | 0.466 | 0.456 | 0.469 | 0.459 | 0.443 | 0.471 | 0.635 | 0.634 | 0.428 | 0.433 | 0.517 | 0.486 |
| ETTm1 → ETTh1 | **0.422** | **0.430** | 0.495 | 0.469 | 0.475 | 0.463 | 0.426 | 0.441 | 0.700 | 0.702 | 0.620 | 0.541 | 0.484 | 0.482 |
| ETTm2 → ETTh1 | **0.428** | **0.441** | 0.464 | 0.456 | 0.453 | 0.450 | 0.503 | 0.507 | 1.091 | 0.814 | 0.598 | 0.548 | 0.616 | 0.550 |
| Weather → ETTh1 | **0.456** | 0.467 | 0.462 | 0.464 | 0.465 | **0.456** | - | - | - | - | 0.518 | 0.487 | 0.463 | 0.460 |
| ETTh1 → ETTm1 | **0.346** | **0.384** | 0.360 | 0.390 | 0.373 | 0.393 | 0.353 | 0.390 | 0.746 | 0.652 | 0.370 | 0.393 | 0.699 | 0.557 |
| ETTh2 → ETTm1 | 0.365 | **0.384** | 0.383 | 0.402 | 0.391 | 0.409 | 0.475 | 0.489 | 0.750 | 0.654 | **0.363** | 0.387 | 0.694 | 0.557 |
| ETTm2 → ETTm1 | **0.351** | **0.383** | 0.390 | 0.410 | 0.382 | 0.402 | 0.414 | 0.464 | 0.758 | 0.699 | 0.385 | 0.412 | 0.423 | 0.420 |
| Weather → ETTm1 | **0.350** | **0.383** | 0.411 | 0.423 | 0.368 | 0.392 | - | - | - | - | 0.382 | 0.403 | 0.382 | 0.395 |
| Avg | **0.392** | **0.413** | 0.429 | 0.434 | 0.422 | 0.428 | 0.436 | 0.460 | 0.780 | 0.693 | 0.458 | 0.451 | 0.535 | 0.488 |

14.7% MSE reduction and 12.0% MAE reduction compared to the contrastive baseline CoST [46]. It is also notable that both Ti-MAE [21] and TST [56] outperform all the contrastive-based baselines. This indicates that masked modeling based on point-wise reconstruction suits the forecasting task better than the series-wise contrastive pre-training.

**Cross-domain.** As demonstrated in Table 3, we present multiple scenarios to verify the effectiveness under the cross-domain setting, where SimMTM consistently outperforms other baselines. Note that the channel-independent encoder enables the comparing baselines to be capable of transferring pre-trained models between datasets with different variate numbers: Weather → {ETTh1, ETTm1}. But for LaST and TF-C with model-specific designs, they cannot be applied to these scenarios. While negative migration has been observed in some cross-domain scenarios, such as Weather → ETTh1 and ETTh2 → ETTm1, SimMTM is still significantly overall superior to other baselines.

Table 4: In- and cross-domain settings of classification. For the in-domain setting, we pre-train and fine-tune the model on the same dataset Epilepsy. For the cross-domain setting, we pre-train a model on SleepEEG and fine-tune it to multiple target datasets: Epilepsy, FD-B, Gesture, EMG. Accuracy (%) score are recorded. Full results are included in Appendix E.

| Models | SimMTM | Random init. | Ti-MAE [21] | TST [56] | LaST [42] | TF-C [57] | CoST [46] | TS2Vec [55] |
|---|---|---|---|---|---|---|---|---|
| Epilepsy → Epilepsy | **94.75** | 89.83 | 80.34 | 80.89 | 92.11 | 93.96 | 92.35 | 92.33 |
| SleepEEG → Epilepsy | **95.49** | 89.83 | 73.45 | 82.89 | 86.46 | 94.95 | 93.66 | 94.46 |
| SleepEEG → FD-B | **69.40** | 47.36 | 67.98 | 65.57 | 46.67 | 69.38 | 54.82 | 60.74 |
| SleepEEG → Gesture | **80.00** | 42.19 | 75.54 | 75.12 | 64.17 | 76.42 | 73.33 | 73.33 |
| SleepEEG → EMG | **97.56** | 77.80 | 63.52 | 75.89 | 66.34 | 81.74 | 73.17 | 80.92 |
| Avg | **87.44** | 69.40 | 72.17 | 76.07 | 71.15 | 83.29 | 77.47 | 80.36 |

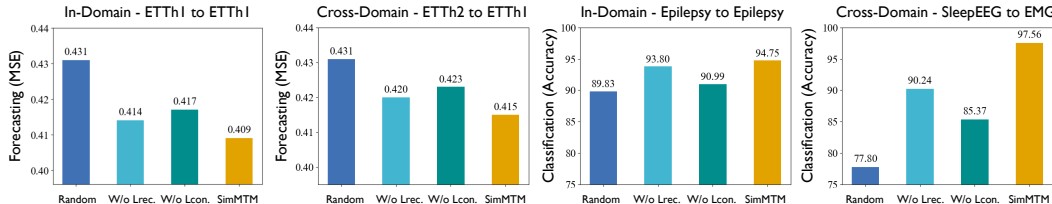

Figure 4: Ablations of SimMTM on the reconstruction loss ($\mathcal{L}_{\text{rec.}}$) and constraint loss ($\mathcal{L}_{\text{con.}}$) in time series forecasting (left part) and classification (right part) tasks under both in-domain and cross-domain settings. More ablations are included in Appendix E.

## 4.3 Classification

**In-domain.** We investigate the in-domain pre-training effect on the classification tasks in Table 4. Note that different from forecasting, the classification task requires the model to learn high-level time series representations. From Table 4, we can find that the contrastive pre-training baselines achieve competitive performances. In contrast, the masking-based model Ti-MAE [21] and TST [56] perform poorly, and TST even exhibits a negative transfer phenomenon compared to the random initialization, indicating that contrastive learning is generally more suitable for classification tasks. Surprisingly, although SimMTM follows the masked modeling paradigm, with our specially-designed reconstruction task, it can still achieve the best performance in the classification task. This is benefited from the neighborhood aggregation from *multiple* masked series, which enables the model to exploit the local structure of the time series manifold.

**Cross-domain.** We experiment with four cross-domain transfer scenarios in Table 4: SleepEEG → {Epilepsy, FD-B, Gesture, EMG}, where the target datasets are distinct from the pre-training dataset. Due to the large gap between pre-training and fine-tuning datasets, the baselines perform poorly in most cases, while SimMTM still surpasses other baselines and the random initialization significantly. Especially for SleepEEG → EMG, SimMTM remarkably surpasses previous state-of-the-art TF-C (*Accuracy*: 97.56% vs. 81.74%). These results demonstrate that SimMTM can precisely capture valuable knowledge from pre-training datasets and uniformly benefit extensive downstream datasets.

## 4.4 Model Analysis

**Ablations.** As shown in Figure 4, we provide ablations to two parts of the training loss in SimMTM. It is observed that both $\mathcal{L}_{\text{reconstruction}}$ and $\mathcal{L}_{\text{constraint}}$ are essential to the final performance. Especially, for the SleepEEG → EMG experiment, SimMTM surpasses the random initialization remarkably, where reconstruction and constraint losses provide 9.7% and 16.0% absolute improvement respectively. Besides, we can also find that in comparison to $\mathcal{L}_{\text{reconstruction}}$, $\mathcal{L}_{\text{constraint}}$ provides more contributions to the final results. This comes from our design that the constraint loss uncovers a proper time series

Table 5: Representation analysis for different methods in classification and forecasting tasks. For each model, we calculate the Centered Kernel Alignment (CKA) similarity (%) [18] between representations from the first and the last encoder layers to measure the representation-learning property of deep models. Since the bottom layer representations usually contain low-level or detailed information, a smaller CKA similarity means the top layer includes different information from the bottom layer and indicates the model tends to learn high-level representations or more abstract information. For comparison, we also calculate the $|\Delta_{\mathrm{CKA}}|$ between pre-trained and fine-tuned models, where a smaller value indicates a smaller representation gap between pre-training and fine-tuning, and the representations have stronger universality and portability.

| Tasks | | Ti-MAE [21] | TST [56] | LaST [42] | TF-C [57] | CoST [46] | TS2vec [55] | SimMTM |
|---|---|---|---|---|---|---|---|---|
| Classification | Pre-training | 84.12 | 54.98 | 82.01 | 85.78 | 60.74 | 70.01 | 33.87 |
| | Fine-tuning | 87.26 | 55.80 | 79.56 | 86.30 | 62.24 | 69.79 | 32.84 |
| | $|\Delta_{\mathrm{CKA}}|$ | 3.14 | 0.82 | 2.45 | 1.53 | 1.50 | **0.22** | 1.04 |
| Forecasting | Pre-training | 83.60 | 99.76 | 75.20 | 59.35 | 87.09 | 70.20 | 97.79 |
| | Fine-tuning | 91.24 | 94.92 | 79.25 | 60.60 | 77.38 | 83.73 | 97.89 |
| | $|\Delta_{\mathrm{CKA}}|$ | 7.64 | 4.84 | 4.05 | 1.25 | 9.70 | 13.53 | **0.11** |
| Sum$|\Delta_{\mathrm{CKA}}|$ | | 10.78 | 5.66 | 6.50 | 2.77 | 11.20 | 13.75 | **1.15** |

Table 6: Performance by applying SimMTM to four advanced time series forecasting models under the in-domain setting. "+ Sub-series Masking" refers to the sub-series masked modeling pre-training method proposed by PatchTST [26] itself. MSE and MAE are averaged from all prediction lengths in {96,192,336,720}. The detailed results for each forecasting horizon are in Appendix E.

| Dataset | ETTh1 | | ETTh2 | | ETTm1 | | ETTm2 | |
|---|---|---|---|---|---|---|---|---|
| Model | MSE | MAE | MSE | MAE | MSE | MAE | MSE | MAE |
| Transformer [39] | 1.088 | 0.836 | 4.103 | 1.612 | 0.901 | 0.704 | 1.624 | 0.901 |
| + SimMTM | **0.927** | **0.761** | **3.498** | **1.487** | **0.809** | **0.663** | **1.322** | **0.808** |
| Autoformer [47] | 0.573 | 0.573 | 0.550 | 0.559 | 0.615 | 0.528 | 0.324 | 0.368 |
| + SimMTM | **0.561** | **0.568** | **0.543** | **0.555** | **0.553** | **0.505** | **0.315** | **0.360** |
| NS Transformer [24] | 0.570 | 0.537 | 0.526 | 0.516 | 0.481 | 0.456 | 0.306 | 0.347 |
| + SimMTM | **0.543** | **0.527** | **0.493** | **0.514** | **0.431** | 0.455 | **0.301** | **0.345** |
| PatchTST [26] | 0.417 | 0.431 | 0.331 | 0.379 | 0.352 | 0.382 | 0.258 | 0.317 |
| + Sub-series Masking | 0.430↓ | 0.445↓ | 0.355↓ | 0.394↓ | **0.341** | 0.379 | 0.258 | 0.318↓ |
| + SimMTM | **0.409** | **0.428** | **0.329** | 0.379 | 0.348 | **0.378** | **0.254** | **0.313** |

manifold helpful for reconstruction from multiple masked series, without which the neighborhood aggregation will degenerate to the trivial average.

**Representation analysis.** To illustrate the advantages of SimMTM intuitively, we provide a representation analysis in Table 5, where we can find the following observations. Firstly, we can find that the CKA value of SimMTM in the classification task is clearly smaller than the values in the forecasting task, where the former is a high-level task and the latter requires low-level representations. These results demonstrate that SimMTM can learn adaptive representations for different tasks, which can be benefited from our design in the pre-training loss. Concretely, the temporal variations of classification pre-training datasets are much more diverse than the forecasting datasets. Thus, the $\mathcal{L}_{\mathrm{constraint}}$ will be easier for optimization in classification, deriving a smaller CKA value. Secondly, from $|\Delta_{\mathrm{CKA}}|$, it is observed that the models pre-trained from SimMTM present a smaller gap with respect to the final fine-tuned model in representation learning properties, which is why SimMTM can consistently improve downstream tasks.

**Model generality.** From Table 6, we can find that as a general time series pre-training framework, SimMTM can consistently improve the forecasting performance of diverse advanced base models, even for the state-of-the-art time series forecasting model PatchTST [26]. This generality also

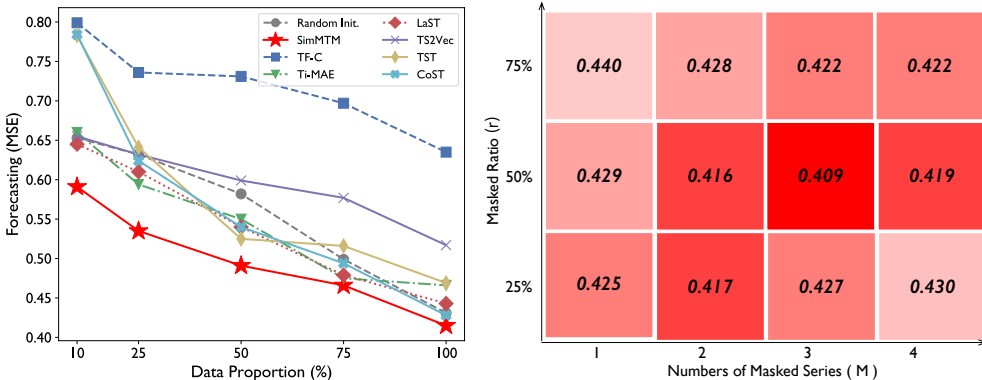

Figure 5: Model analysis. Left part is for fine-tuning ETTh2 pre-trained model to ETTh1 with limited data, where a smaller MSE indicates better performance. Right part presents the MSE performance of SimMTM in the ETTh1 "input-336-predict-96" in-domain setting with different masked ratio $r$ and numbers of masked series $M$, where a darker red means better performance.

indicates that we can further improve the model's performance by employing advanced base models as encoders. It is also notable that different from the negative transfer phenomenon caused by the canonical sub-series masked modeling used in the PatchTST paper [26], the consistency improvement of SimMTM further verifies the effectiveness of our design.

**Fine-tuning to limited data scenarios.** One essential application of pre-training models is to provide prior knowledge for downstream tasks, especially for limited data scenarios, which is critical to the fast-adaption of deep models. Thus, to verify the effectiveness of SimMTM and other pre-training methods in data-limited scenarios, we pre-train a model on ETTh2 and fine-tune it to ETTh1 with different choices for the remaining proportions of training data. All results are presented in Figure 5. We can find that SimMTM achieves significant performance gains in different data proportions compared to other time series pre-training methods. Specifically, for the 10% data fine-tuning setting, SimMTM significantly outperforms the advanced masking-based method Ti-MAE [21] (*MSE*: 0.591 vs. 0.660). Compared with the contrastive-based method TF-C [57], SimMTM also achieves 26.0% MSE reduction. These results further verify that SimMTM can effectively capture valuable knowledge from datasets and boost the final performance, even in limited data scenarios.

**Masking strategy.** Note that the difficulty of reconstructing the original time series increases along with the increase of the masked ratio, but decreases when the number of neighbor masked series increases. We explore the potential relationship between the masked ratio and the number of masked series used for reconstruction, namely $r$ and $M$ in Eq. (1) respectively. The experimental results in Figure 5 show that we need to set $M \propto r$ to obtain better results. Experimentally, we choose the masking ratio as 50% and adopt three masked series for reconstruction throughout this paper. In addtion, we can find that only using one masked series ($M = 1$) for pre-training generally performs worse than the settings with larger $M$, where the latter enables the model to discover the relations between input series and its neighbors. See Figure 1 for an intuitive understanding. These results further highlight the advantage of our design in manifold learning.

## 5  Conclusion

This paper presents SimMTM, a simple pre-training framework for masked time-series modeling. Going beyond the previous convention in reconstructing the original time series from unmasked time points, SimMTM proposes a new masked modeling task as reconstructing the original series from its multiple neighbor masked series. Concretely, SimMTM aggregates the point-wise representations based on the series-wise similarities, which are carefully constrained by the neighborhood assumption on the time series manifold. Experimentally, SimMTM can furthest bridge the gap between pre-trained and fine-tuned models, thereby achieving consistent state-of-the-art in distinct forecasting and classification tasks compared to the most advanced time series pre-training methods, covering both in-domain and cross-domain settings. In the future, we will further extend SimMTM to large-scale and diverse pre-training datasets in pursuing the foundation model for time series analysis.

## Acknowledgments

This work was supported by the National Key Research and Development Plan (2021YFB1715200), National Natural Science Foundation of China (62022050 and 62021002), and Beijing Nova Program (Z201100006820041).

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

# A  Implementation Details

All the experiments are repeated five times, implemented in PyTorch [28] and conducted on NVIDIA A100 SXM4 40GB GPU. We implement the baselines based on their official implementation and follow the configuration from their original papers. For the metrics, we adopt the mean square error (MSE) and mean absolute error (MAE) for the time series forecasting. As for the classification, accuracy, precision, recall, F1 score, and their average value are recorded.

## A.1  Dataset Description

We conduct experiments to evaluate the effect of our method under in-domain and cross-domain settings on twelve real-world datasets for two typical time series analysis tasks: forecasting and classification, covering diverse application scenarios (electricity system, neurological healthcare, human activity recognition, mechanical fault detection, and physical status monitoring), different types of signals (ECG, EMG, acceleration, vibration, power load, weather, and transoirtation), multivariate channel dimensions (from 1 to 862), varying times series lengths (from 96 to 5120) and large span sampling ratio (from 100 Hz to 4000 Hz). The detailed descriptions of these datasets are summarized in Table 7.

Table 7: Dataset descriptions. *Samples* are organized in (Train/Validation/Test).

| Tasks | Datasets | Channels | Length | Samples | Classes | Information | Frequency |
|---|---|---|---|---|---|---|---|
| Forecasting | ETTh1,ETTh2 | 7 | {96,192,336,720} | 8545/2881/2881 | - | Electricity | 1 Hour |
| | ETTm1,ETTm2 | 7 | {96,192,336,720} | 34465/11521/11521 | - | Electricity | 15 Mins |
| | Weather | 21 | {96,192,336,720} | 36792/5271/10540 | - | Weather | 10 Mins |
| | Electricity | 321 | {96,192,336,720} | 18317/2633/5261 | - | Electricity | 1 Hour |
| | Traffic | 862 | {96,192,336,720} | 12185/1757/3509 | - | Transportation | 1 Hour |
| Classification | SleepEEG | 1 | 200 | 371005/-/- | 5 | EEG | 100 Hz |
| | Epilepsy | 1 | 178 | 60/20/11420 | 2 | EEG | 174 Hz |
| | FD-B | 1 | 5120 | 60/21/135599 | 3 | Faulty Detection | 64K Hz |
| | Gesture | 3 | 315 | 320/120/120 | 8 | Hand Movement | 100 Hz |
| | EMG | 1 | 1500 | 122/41/41 | 3 | Muscle responses | 4K Hz |

(1) **ETT (4 subsets)** [58] contains the time series of oil temperature and power load collected by electricity transformers from July 2016 to July 2018. ETT is a group of four subsets with different recorded frequencies: ETTh1/ETTh2 are recorded every hour, and ETTm1/ETTm2 are recorded every 15 minutes.

(2) **WEATHER** [43] includes meteorological time series with 21 weather indicators collected every 10 minutes from the Weather Station of the Max Planck Biogeochemistry Institute in 2020.

(3) **ELECTRICITY** [38] records the hourly electricity consumption of 321 clients from 2012 to 2014. Values are in kW of each 15 min. All time labels report to Portuguese hour. However, all days present 96 measures (24×4). For every year in March, time change day (which has only 23 hours), values between 1:00 am and 2:00 am are zero for all points. For every year in October, time change day (which has 25 hours), the values between 1:00 am and 2:00 am aggregated consumption of two hours.

(4) **TRAFFIC** [29] encompasses the hourly measures of road occupancy rates obtained from 862 sensors situated in the San Francisco Bay area freeways. These measurements were carried out between January 2015 and December 2016.

(5) **SLEEPEEG** [16] contains 153 whole-night sleeping electroencephalography (EEG) recordings from 82 healthy subjects. We follow the same data preprocessing approach as [57] and get 371,055 univariate brainwaves. Each brainwave is sampled at a frequency of 100 Hz and associated with one of five sleeping stages: Wake, Non-rapid eye movement (3 sub-states), and Rapid Eye Movement.

(6) **EPILEPSY** [1] monitors the brain activities of 500 subjects with a single-channel EEG sensor. Every subject is recorded for 23.6 seconds of brain activities. The dataset is sampled at 178 Hz and contains 11,500 samples in total. We follow the procedure described by [57]. The first four classes (eyes open, eyes closed, EEG measured in the healthy brain region, and EEG measured in the tumor region) of the original five categories of each sample are classified as positive, and the remaining classes are used as negative.

(7) **FD-B** [19] is generated by electromechanical drive systems. It monitors the condition of rolling bearings and detects their failures based on the monitoring conditions, which include speed, load torque, and radial force. Concretely, FD-B has 13,640 samples in total. Each recording is sampled at 64k Hz with 3-class labels: undamaged, inner damaged, and outer damaged.

(8) **GESTURE** [22] are collected from 8 hand gestures based on the paths of hand movement recorded by an accelerometer. The eight gestures are hand swiping left, right, up, and down, hand waving in a counterclockwise or clockwise circle, hand waving in a square, and waving a right arrow, respectively. This dataset contains 440 examples of balanced classification labels that can be used, and each sample includes eight different kinds of gesture categories.

(9) **EMG** [30] is sampled with 4K Hz and consists of 163 single-channel EMG recordings from the anterior tibialis muscle of three healthy volunteers suffering from neuropathy and myopathy. Each patient is a classification category, so each sample is associated with one of three classes.

## A.2 Baselines Implementation

We compare SimMTM against six state-of-the-art baselines. To make a fair and comprehensive comparison, we tried two baseline implementation approaches for forecasting and classification tasks: the unified encoder and reproduced with their official implementation encoder. Notably, LaST [42] and TF-C [57] are closely related to model structures. We directly report results from their papers or reproduce codes with official implementation.

Table 8: Baselines implementation details.

| Baselines | Task | Encoder | Performance Comparison | Report |
|---|---|---|---|---|
| Ti-MAE [21] | Forecasting | Channel-independent Transformer | better | Main text |
| | | Official implementation | | Section E |
| | Classification | 1D-ResNet | better | Main text |
| | | Official implementation | | Section E |
| TST [56] | Forecasting | Channel-independent Transformer | better | Main text |
| | | Official implementation | | Section E |
| | Classification | 1D-ResNet | better | Main text |
| | | Official implementation | | Section E |
| LaST [42] | Forecasting | Official implementation | / | Main text |
| | Classification | Official implementation | / | Main text |
| TF-C [57] | Forecasting | Official implementation | / | Main text |
| | Classification | Official implementation | / | Main text |
| CoST [46] | Forecasting | Channel-independent Transformer | better | Main text |
| | | Official implementation | | Section E |
| | Classification | 1D-ResNet | better | Main text |
| | | Official implementation | | Section E |
| TS2Vec [55] | Forecasting | Channel-independent Transformer | better | Main text |
| | | Official implementation | | Section E |
| | Classification | 1D-ResNet | better | Main text |
| | | Official implementation | | Section E |

(1) **Unified encoder**. We attempt to unify the encoder for these pre-training methods. Specifically, we adopt the vanilla Transformer [39] with channel independent [26] for forecasting to accomplish cross-domain transfer between datasets with different variate numbers. As for the classification, we use 1D-ResNet [13] as the

encoder following [57]. Besides, we do a comprehensive hyperparameter search for all baselines. For the Transformer encoder, we vary the number of Transformer layers in $\{1, 2, 3, 4\}$, select the model dimension from $\{16, 32, 64, 128, 256\}$, and the attention head from $\{4, 8, 16, 32\}$. For the 1D-ResNet, we search the number of 1D-ResNet layers from $\{1, 2, 3, 4\}$, the kernel size from $\{3, 5, 8\}$ respectively. Additionally, for the masked modeling methods TST [56], Ti-MAE [21], we also searched the masked ratio $r = \{0.125, 0.25, 0.5, 0.75\}$ for better performance.

(2) **Official implementation**. We also implement the baselines following the corresponding official codes, including encoder, hyperparameters, etc. The comparisons are included in Section E of this supplementary material. We directly report the results from their original papers for the same set. For mismatched settings, the results are from our implementation.

Finally, for baselines Ti-MAE [21], TST [56], CoST [46], and TS2Vec [55], we report the results based on the unified encoder in the main text. But for baselines LaST [42] and TF-C [57], we report the results of the official code implementation or their original paper, which are limited by their model structures. As a result, the performances of all baselines with unified encoder (that we reported in the main text) generally surpass their official implementation and results reported in their own paper. Table 8 shows more details. Full experimental results are in Section E.

## A.3 Pre-training and Fine-tuning Configuration

We built two types of pre-training and fine-tuning scenarios, in-domain and cross-domain, based on the benchmarks of forecasting and classification tasks to compare the effectiveness of our method and other time series pre-training methods.

We pre-train a model on one subset for forecasting tasks and fine-tune it to the same dataset to build seven in-domain transfer evaluation scenarios. In cross-domain evaluation, we pre-train a model on one specific dataset and use other datasets for fine-tuning. Based on the above settings, we constructed fifteen in- and cross-domain pre-training and fine-tuning experiments, covering the same dataset with the same sampled frequency, different datasets with the same sampled frequency, and different datasets with different sampled frequencies.

We use the same dataset, Epilepsy, to construct the in-domain setting for classification tasks. For the cross-domain setting, we pre-train a model for classification tasks on a univariate time series dataset SleepEEG with the most complex temporal dynamics and the most samples. And then fine-tune the model separately on Epilepsy, FD-B, Gesture, and EMG. Furthermore, we constructed four cross-domain evaluation scenarios by pre-training from SleepEEG and fine-tuning to Epilepsy, FD-B, Gesture, and EMG because of fewer commonalities and the enormous gap among these datasets. Table 9 shows detailed pre-training and fine-tuning settings.

Table 9: Pre-training and fine-tuning scenarios for time series forecasting (Fore.) and classification (Class.) tasks, including the same and different datasets and in- and cross-domain settings.

| Tasks | Evaluation | Scenarios | Characteristic |
|---|---|---|---|
| Fore. | In-domain | ETTh1 $\rightarrow$ ETTh1
ETTh2 $\rightarrow$ ETTh2
ETTm1 $\rightarrow$ ETTm1
ETTm2 $\rightarrow$ ETTm2
Weather $\rightarrow$ Weather
Electricity $\rightarrow$ Electricity
Traffic $\rightarrow$ Traffic | The same dataset with the same frequency |
| | Cross-domain | ETTh2 $\rightarrow$ ETTh1
ETTm2 $\rightarrow$ ETTm1 | Different datasets with the same frequency. |
| | | {ETTm1, ETTm2, Weather} $\rightarrow$ ETTh1
{ETTh1, ETTh2, Weather} $\rightarrow$ ETTm1 | Different datasets with different frequencies. |
| Class. | In-domain | Epilepsy $\rightarrow$ Epilepsy | The same dataset with the same frequency. |
| | Cross-domain | SleepEEG $\rightarrow$ {Epilepsy, FD-B, Gesture, EMG} | Different datasets with different frequencies. |

## A.4 Model and Training Configuration

Following the previous convention, we choose the encoder part of Transformer [39] with channel independent as the feature extractor for forecasting tasks. For the classification tasks, we adopt 1D-ResNet [13] as the encoder following [57]. In the pre-training stages, we pre-train the model with different learning rates and batch sizes according to the pre-train datasets. Then we fine-tune it to downstream forecasting and classification tasks supervised by L2 and Cross-Entropy losses, respectively. The configuration details are in Table 10.

Table 10: Model and training configuration in Forecasting (*Fore.*) and Classification (*Class.*) tasks.

| Tasks | Encoder | | Pre-training | | | Fine-tuning | | | |
|---|---|---|---|---|---|---|---|---|---|
| | $e_{\text{layers}}$ | $d_{\text{model}}$ | learning rate | batch size | epochs | learning rate | loss function | batch size | epochs |
| Fore. | 2 | 16 | 1e-3 | 32 | 50 | 1e-4 | L2 | {16,32} | 10 |
| Class. | 3 | 128 | 1e-4 | 128 | 10 | 1e-4 | Cross-Entropy | 32 | 300 |

## B Hyperparameter Sensitivity

We verify the hyperparameter sensitivity of SimMTM on ETTh1 in Table 11, including masked ratio (r), the number of masked series (M), temperature ($\tau$), masked function ($\text{Mask}$), encoder depth ($e_{\text{layers}}$), and the hidden dimension ($d_{\text{model}}$). Lower MSE and MAE represent better performance.

As shown in Table 11 (a) and 11 (b), we can observe the effect of the method is closely related to the trade-off of the masked ratio and the number of masked series. Hence, a reasonable balance between the two kinds of parameters is critical. For the temperature hyperparameter of softmax normalization ($\tau$), we use an appropriately small $\tau$ that leads to higher differences and diversity of masked sequences. For the masked methods, we chose two masked methods for verification: masking following random distribution and masking following geometric distribution [56]. The results show that the method based on geometric masking is better than random masking modeling. Besides, we can find that 2 encoder layers are enough for reconstruction tasks. Note our method SimMTM consistently performs better than training from scratch under various hyperparameter changes.

Table 11: Hyperparameter sensitivity experiments on ETTh1 for the in-domain setting. The entries marked in bold are the same which specify the default settings. This table format follows [11].

| (a) Masked ratio | | | (b) Masked numbers | | | (c) Temperature | | |
|---|---|---|---|---|---|---|---|---|
| Ratio | MSE | MAE | Numbers | MSE | MAE | Value | MSE | MAE |
| 12.5% | 0.429 | 0.440 | 1 | 0.429 | 0.437 | 0.02 | **0.409** | **0.428** |
| 25% | 0.427 | 0.434 | 2 | 0.416 | 0.429 | 0.2 | **0.409** | 0.429 |
| 50% | **0.409** | **0.428** | 3 | **0.409** | **0.428** | 2 | 0.416 | 0.428 |
| 75% | 0.422 | 0.434 | 4 | 0.419 | 0.431 | | | |

| (d) Masked function | | | (e) Encoder depth | | | (f) Hidden layer dimension | | |
|---|---|---|---|---|---|---|---|---|
| Type | MSE | MAE | Layers | MSE | MAE | Dim | MSE | MAE |
| Random | 0.409 | 0.431 | 1 | 0.420 | 0.426 | 16 | **0.409** | **0.428** |
| Geometric | **0.409** | **0.428** | 2 | **0.409** | **0.428** | 32 | 0.420 | 0.432 |
| | | | 3 | 0.421 | 0.430 | 64 | 0.422 | 0.434 |
| | | | 4 | 0.426 | 0.436 | 128 | 0.428 | 0.444 |

# C   Ablations on Aggregation Setting

SimMTM proposes to recover masked time points by the weighted aggregation of multiple neighbors outside the manifold. We explored two types of aggregation settings.

(1) **Positive Samples Aggregation (PSA)**: only aggregate multiple positive neighbors (the masked series of the same sample) to reconstruct masked time points.

(2) **Positive and Negative Samples Aggregation (PNSA)**: aggregate both positive and negative neighbors (the masked series of all samples) to reconstruct masked time points.

As shown in Table 12, although PSA made good progress compared to training from scratch (Random Init.), PNSA is consistently better than SimMTM PSA in all ablation settings. In masked time-series modeling, masking can be viewed as adding noise to the original data, and masked modeling is to project masked data from the neighborhood back to the original manifold. We use positive and negative masked time series as the reconstruction candidates to drive the model to select the positive samples adaptively, which can make the model learn the structure of the manifold better. Therefore, as stated in the Method Section of the main text, we choose positive and negative sample aggregation (PNSA) as the standard aggregation setting of SimMTM.

Table 12: Ablations on aggregation setting in forecasting (*MSE*) and classification (*Acc*) tasks for in- and cross-domain settings. A smaller MSE or a higher Accuracy indicates a better result (↑).

| Tasks | Evaluation | Scenarios | Aggregation | Metric |
|---|---|---|---|---|
| Forecasting | In-domain | ETTh1 → ETTh1 | Random init. | 0.431 |
| | | | SimMTM (**PSA**) | 0.420 ↑ |
| | | | SimMTM (**PNSA**) | **0.409** ↑ |
| | Cross-domain | ETTh2 → ETTh1 | Random init. | 0.431 |
| | | | SimMTM (**PSA**) | 0.426 ↑ |
| | | | SimMTM (**PNSA**) | **0.415** ↑ |
| Classification | In-domain | Epilepsy → Epilepsy | Random init. | 89.83 |
| | | | SimMTM (**PSA**) | 92.56 ↑ |
| | | | SimMTM (**PNSA**) | **94.75** ↑ |
| | Cross-domain | SleepEEG → EMG | Random init. | 77.80 |
| | | | SimMTM (**PSA**) | 87.80 ↑ |
| | | | SimMTM (**PNSA**) | **97.56** ↑ |

# D   Comparison of Masked Modeling

To investigate the reconstruction process of different masked modeling methods, we plot both original and reconstructed time series from TST and SimMTM in Figure 6, where TST [56] follows the canonical masked modeling paradigm and learns to predict removed time points based on the remaining time points. In Figure 6, we can find that direct reconstruction is too difficult in time series, even for the 12.5% masking ratio. As for the 75% masking ratio, TST degenerates more seriously. Because of this poor reconstruction effect, direct reconstruction is difficult to provide reliable guidance to model pre-training. In contrast, our proposed SimMTM can precisely reconstruct the original time series, benefiting the representation learning. These results also support our design in neighborhood reconstruction.

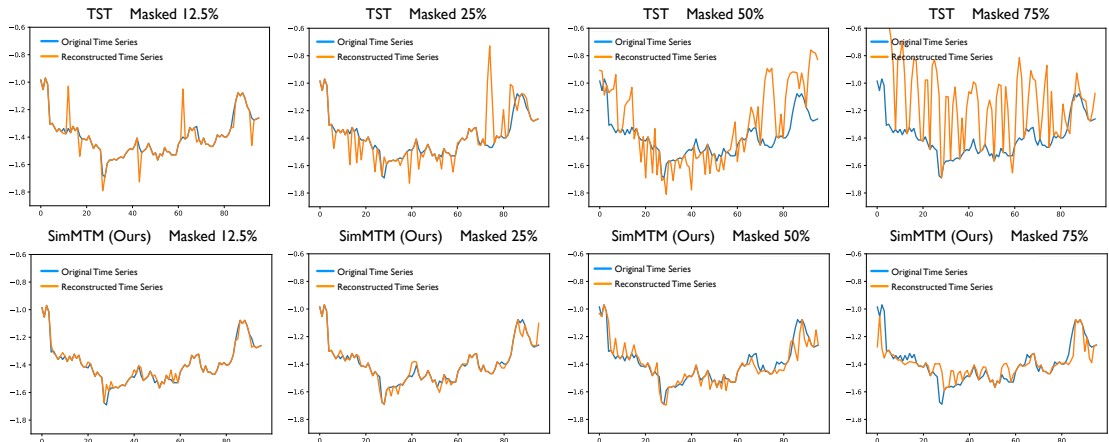

Figure 6: Comparison of the canonical masked modeling paradigm TST and neighborhood aggregation masked modeling SimMTM in reconstructing time series. All the cases are shown from ETTh1.

# E    Full Results

Due to the limited length of the text, we summarize all the experiments in the main text into two parts: the main experiment and the analytical experiment. We categorize and index them in Tabel 13, 14.

Table 13: The main results of pre-training and fine-tuning scenarios for time series forecasting and classification tasks, including the same and different encoder for in- and cross-domain settings.

| Tasks | Evaluation | Encoder | Tabels Name |
|---|---|---|---|
| Forecasting | In-domain | The model utilized in the original papers | Table 15 |
| | | Transformer with channel independent | Table 16 |
| | Cross-domain | The model utilized in the original papers | Table 17 |
| | | Transformer with channel independent | Table 18 |
| Classification | In-domain | The model utilized in the original papers | Table 23 |
| | | 1D-ResNet | Table 24 |
| | Cross-domain | The model utilized in the original papers | Table 23 |
| | | 1D-ResNet | Table 24 |

Table 14: The model analysis results of pre-training and fine-tuning scenarios for time series forecasting and classification tasks with the unified encoder for in- and cross-domain settings.

| Tasks | Evaluation | Analysis | Tabels Name |
|---|---|---|---|
| Forecasting | In-domain | Ablation study | Table 19 |
| | | Model generality | Table 21 |
| | Cross-domain | Ablation study | Table 20 |
| | | Limited data | Table 22 |
| Classification | In-domain | Ablation study | Table 25 |
| | Cross-domain | Ablation study | Table 25 |

## F    Limitations

SimMTM is inspired by the manifold perspective of masked modeling. Although we have provided relatively comprehensive results to verify the model's effectiveness, the model performance still needs theoretical guarantees. In fact, the most high-impact works in the self-supervised pre-training community are without theoretical analysis, such as BERT [7], GPT-3 [10], MAE [11] and SimMIM [50]. Thus, we would like to leave this problem as a future work.

The masking ratio of masked modeling methods is an essential hyper-parameter. Although we have provided a chosen principle to masking ratio $r$ and the number of masked time series $M$ as $M \propto r$ in the main text, we still need to tune these two hyperparameters for different datasets to achieve the best performance. Notably, previous methods also chose the masking ratio solely based on the empirical results [7, 11]. Thus, despite there exist limitations of SimMTM in choosing hyperparameters, the principle of $M \propto r$ can somewhat ease this problem. And the chosen strategy of the masking ratio can also be a potential topic in masked modeling [44].

## G    Social Impacts

This paper presents SimMTM as a new masked modeling method for time series. SimMTM achieves state-of-the-art in two mainstream time series analysis tasks, which can be a good supplement for the self-supervised pre-training community. We will also publish the codebase of time-series pre-training to facilitate future research.

This paper only focuses on the algorithm design. Using all the codes and datasets strictly follows the corresponding licenses (Appendix A.1). There is no potential ethical risk or negative social impact.

Table 15: Complete results of long-term forecasting tasks for the in-domain setting of forecasting the future $O \in \{96, 192, 336, 720\}$ time points based on the past 336 time points. **All the results of baselines are based on the encoder utilized in their original papers.** The standard deviations of SimMTM are within 0.005 for MSE and within 0.004 for MAE.

| Models | | SimMTM | | Random init. | | Ti-MAE [21] | | TST [56] | | LaST [42] | | TF-C [57] | | CoST [46] | | TS2Vec [55] | |
|---|---|---|---|---|---|---|---|---|---|---|---|---|---|---|---|---|---|
| Metric | | MSE | MAE | MSE | MAE | MSE | MAE | MSE | MAE | MSE | MAE | MSE | MAE | MSE | MAE | MSE | MAE |
| ETTh1 | 96 | **0.379** | **0.407** | 0.380 | 0.412 | 0.708 | 0.570 | 0.503 | 0.527 | 0.399 | 0.412 | 0.463 | 0.406 | 0.514 | 0.512 | 0.709 | 0.650 |
| | 192 | **0.412** | **0.424** | 0.416 | 0.434 | 0.725 | 0.587 | 0.601 | 0.552 | 0.484 | 0.468 | 0.531 | 0.540 | 0.655 | 0.590 | 0.927 | 0.757 |
| | 336 | **0.421** | **0.431** | 0.448 | 0.458 | 0.713 | 0.589 | 0.625 | 0.541 | 0.580 | 0.533 | 0.535 | 0.545 | 0.790 | 0.666 | 0.986 | 0.811 |
| | 720 | **0.424** | 0.449 | 0.481 | 0.487 | 0.736 | 0.618 | 0.768 | 0.628 | 0.432 | **0.432** | 0.577 | 0.562 | 0.880 | 0.739 | 0.967 | 0.790 |
| | Avg | **0.409** | **0.428** | 0.431 | 0.448 | 0.721 | 0.591 | 0.624 | 0.562 | 0.474 | 0.461 | 0.527 | 0.513 | 0.710 | 0.627 | 0.897 | 0.752 |
| ETTh2 | 96 | **0.293** | **0.347** | 0.325 | 0.374 | 0.443 | 0.465 | 0.335 | 0.392 | 0.331 | 0.390 | 0.463 | 0.521 | 0.465 | 0.482 | 0.506 | 0.477 |
| | 192 | **0.355** | **0.386** | 0.400 | 0.424 | 0.533 | 0.516 | 0.444 | 0.441 | 0.451 | 0.452 | 0.525 | 0.561 | 0.671 | 0.599 | 0.567 | 0.547 |
| | 336 | **0.370** | **0.401** | 0.405 | 0.433 | 0.445 | 0.472 | 0.455 | 0.494 | 0.460 | 0.478 | 0.850 | 0.883 | 0.848 | 0.776 | 0.694 | 0.628 |
| | 720 | **0.395** | **0.427** | 0.451 | 0.475 | 0.507 | 0.498 | 0.481 | 0.504 | 0.552 | 0.509 | 0.930 | 0.932 | 0.871 | 0.811 | 0.728 | 0.838 |
| | Avg | **0.353** | **0.390** | 0.395 | 0.427 | 0.482 | 0.488 | 0.429 | 0.458 | 0.449 | 0.457 | 0.692 | 0.724 | 0.714 | 0.667 | 0.624 | 0.623 |
| ETTm1 | 96 | **0.288** | 0.348 | 0.295 | **0.346** | 0.647 | 0.497 | 0.454 | 0.456 | 0.316 | 0.355 | 0.419 | 0.401 | 0.376 | 0.420 | 0.563 | 0.551 |
| | 192 | **0.327** | 0.373 | 0.333 | 0.374 | 0.597 | 0.508 | 0.471 | 0.490 | 0.349 | **0.366** | 0.471 | 0.438 | 0.420 | 0.451 | 0.599 | 0.558 |
| | 336 | **0.363** | **0.395** | 0.370 | 0.398 | 0.699 | 0.525 | 0.457 | 0.451 | 0.429 | 0.407 | 0.540 | 0.509 | 0.482 | 0.494 | 0.685 | 0.594 |
| | 720 | **0.412** | **0.424** | 0.427 | 0.431 | 0.786 | 0.596 | 0.594 | 0.488 | 0.496 | 0.464 | 0.552 | 0.548 | 0.628 | 0.578 | 0.831 | 0.698 |
| | Avg | **0.348** | **0.385** | 0.356 | 0.387 | 0.682 | 0.532 | 0.494 | 0.471 | 0.398 | 0.398 | 0.496 | 0.474 | 0.477 | 0.486 | 0.670 | 0.600 |
| ETTm2 | 96 | 0.172 | 0.261 | 0.175 | 0.268 | 0.304 | 0.357 | 0.363 | 0.301 | **0.163** | **0.255** | 0.401 | 0.477 | 0.276 | 0.384 | 0.448 | 0.482 |
| | 192 | **0.223** | **0.300** | 0.240 | 0.312 | 0.334 | 0.387 | 0.342 | 0.364 | 0.239 | 0.303 | 0.422 | 0.490 | 0.500 | 0.532 | 0.545 | 0.536 |
| | 336 | 0.282 | **0.331** | 0.298 | 0.351 | 0.420 | 0.441 | 0.414 | 0.361 | **0.259** | 0.366 | 0.513 | 0.508 | 0.680 | 0.695 | 0.681 | 0.744 |
| | 720 | **0.374** | 0.388 | 0.403 | 0.413 | 0.508 | 0.481 | 0.580 | 0.456 | 0.397 | **0.382** | 0.523 | 0.772 | 0.925 | 0.914 | 0.691 | 0.837 |
| | Avg | **0.263** | **0.320** | 0.279 | 0.336 | 0.392 | 0.417 | 0.425 | 0.371 | 0.265 | 0.327 | 0.465 | 0.562 | 0.595 | 0.631 | 0.591 | 0.650 |
| Weather | 96 | 0.158 | **0.211** | 0.166 | 0.216 | 0.216 | 0.280 | 0.292 | 0.370 | **0.153** | **0.211** | 0.215 | 0.296 | 0.327 | 0.359 | 0.433 | 0.462 |
| | 192 | **0.199** | **0.249** | 0.208 | 0.254 | 0.303 | 0.335 | 0.410 | 0.473 | 0.207 | 0.250 | 0.267 | 0.345 | 0.390 | 0.422 | 0.508 | 0.518 |
| | 336 | **0.246** | 0.286 | 0.257 | 0.290 | 0.351 | 0.358 | 0.434 | 0.427 | 0.249 | **0.264** | 0.299 | 0.360 | 0.477 | 0.446 | 0.545 | 0.549 |
| | 720 | **0.317** | 0.337 | 0.326 | 0.338 | 0.425 | 0.399 | 0.539 | 0.523 | 0.319 | **0.320** | 0.361 | 0.395 | 0.551 | 0.586 | 0.576 | 0.572 |
| | Avg | **0.230** | 0.271 | 0.239 | 0.275 | 0.324 | 0.343 | 0.419 | 0.448 | 0.232 | **0.261** | 0.286 | 0.349 | 0.436 | 0.453 | 0.516 | 0.525 |
| Electricity | 96 | **0.133** | **0.223** | 0.190 | 0.279 | 0.399 | 0.412 | 0.292 | 0.370 | 0.166 | 0.254 | 0.366 | 0.436 | 0.230 | 0.353 | 0.322 | 0.401 |
| | 192 | **0.147** | **0.237** | 0.195 | 0.285 | 0.400 | 0.460 | 0.270 | 0.373 | 0.178 | 0.278 | 0.366 | 0.433 | 0.253 | 0.371 | 0.343 | 0.416 |
| | 336 | **0.166** | **0.265** | 0.211 | 0.301 | 0.564 | 0.573 | 0.334 | 0.323 | 0.186 | 0.275 | 0.358 | 0.428 | 0.197 | 0.287 | 0.362 | 0.435 |
| | 720 | **0.203** | 0.297 | 0.253 | 0.333 | 0.880 | 0.770 | 0.344 | 0.346 | 0.213 | **0.288** | 0.363 | 0.431 | 0.230 | 0.328 | 0.388 | 0.456 |
| | Avg | **0.162** | **0.256** | 0.212 | 0.300 | 0.561 | 0.554 | 0.310 | 0.353 | 0.186 | 0.274 | 0.363 | 0.432 | 0.228 | 0.335 | 0.354 | 0.427 |
| Traffic | 96 | **0.368** | **0.262** | 0.471 | 0.309 | 0.431 | 0.482 | 0.559 | 0.454 | 0.706 | 0.385 | 0.613 | 0.340 | 0.751 | 0.431 | 0.321 | 0.367 |
| | 192 | **0.373** | **0.251** | 0.475 | 0.308 | 0.491 | 0.346 | 0.583 | 0.493 | 0.709 | 0.388 | 0.619 | 0.516 | 0.751 | 0.424 | 0.476 | 0.367 |
| | 336 | **0.395** | **0.254** | 0.490 | 0.315 | 0.502 | 0.384 | 0.637 | 0.469 | 0.714 | 0.394 | 0.785 | 0.497 | 0.761 | 0.425 | 0.499 | 0.376 |
| | 720 | **0.432** | **0.290** | 0.524 | 0.332 | 0.533 | 0.543 | 0.663 | 0.594 | 0.723 | 0.421 | 0.850 | 0.472 | 0.780 | 0.433 | 0.563 | 0.390 |
| | Avg | **0.392** | **0.264** | 0.490 | 0.316 | 0.489 | 0.399 | 0.611 | 0.503 | 0.713 | 0.397 | 0.717 | 0.456 | 0.761 | 0.428 | 0.501 | 0.375 |

Table 16: Complete results of long-term forecasting tasks for the in-domain setting of forecasting the future $O \in \{96, 192, 336, 720\}$ time points based on the past 336 time points. **All the results of baseline are based on the unified channel-independent Transformer encoder.** The standard deviations of SimMTM are within 0.005 for MSE and within 0.004 for MAE.

| Models | | **SimMTM** | | Random init. | | Ti-MAE [21] | | TST [56] | | LaST [42] | | TF-C [57] | | CoST [46] | | TS2Vec [55] | |
|---|---|---|---|---|---|---|---|---|---|---|---|---|---|---|---|---|---|
| Metric | | MSE | MAE | MSE | MAE | MSE | MAE | MSE | MAE | MSE | MAE | MSE | MAE | MSE | MAE | MSE | MAE |
| ETTh1 | 96 | 0.379 | **0.407** | 0.380 | 0.412 | **0.356** | 0.420 | 0.401 | 0.425 | - | - | - | - | 0.422 | 0.436 | 0.392 | 0.420 |
| | 192 | **0.412** | **0.424** | 0.416 | 0.434 | 0.421 | 0.434 | 0.427 | 0.432 | - | - | - | - | 0.520 | 0.487 | 0.445 | 0.452 |
| | 336 | **0.421** | **0.431** | 0.448 | 0.458 | 0.447 | 0.446 | 0.519 | 0.487 | - | - | - | - | 0.472 | 0.462 | 0.453 | 0.455 |
| | 720 | **0.424** | **0.449** | 0.481 | 0.487 | 0.469 | 0.482 | 0.515 | 0.504 | - | - | - | - | 0.525 | 0.501 | 0.495 | 0.496 |
| | Avg | **0.409** | **0.428** | 0.431 | 0.448 | 0.423 | 0.446 | 0.466 | 0.462 | - | - | - | - | 0.485 | 0.472 | 0.446 | 0.456 |
| ETTh2 | 96 | **0.293** | **0.347** | 0.325 | 0.374 | 0.339 | 0.378 | 0.322 | 0.358 | - | - | - | - | 0.321 | 0.374 | 0.365 | 0.509 |
| | 192 | **0.355** | **0.386** | 0.400 | 0.424 | 0.380 | 0.402 | 0.448 | 0.435 | - | - | - | - | 0.380 | 0.403 | 0.396 | 0.422 |
| | 336 | **0.370** | **0.401** | 0.405 | 0.433 | 0.388 | 0.323 | 0.420 | 0.440 | - | - | - | - | 0.430 | 0.451 | 0.399 | 0.436 |
| | 720 | **0.395** | **0.427** | 0.451 | 0.475 | 0.414 | 0.442 | 0.424 | 0.452 | - | - | - | - | 0.466 | 0.480 | 0.508 | 0.503 |
| | Avg | **0.353** | **0.390** | 0.395 | 0.427 | 0.380 | 0.386 | 0.404 | 0.421 | - | - | - | - | 0.399 | 0.427 | 0.417 | 0.468 |
| ETTm1 | 96 | **0.288** | 0.348 | 0.295 | 0.346 | 0.305 | 0.351 | 0.310 | 0.348 | - | - | - | - | 0.291 | **0.343** | 0.681 | 0.689 |
| | 192 | **0.327** | 0.373 | 0.333 | 0.374 | 0.343 | 0.374 | 0.362 | 0.380 | - | - | - | - | 0.330 | **0.370** | 0.689 | 0.551 |
| | 336 | **0.363** | **0.395** | 0.370 | 0.398 | 0.387 | 0.407 | 0.389 | 0.402 | - | - | - | - | 0.382 | 0.401 | 0.704 | 0.559 |
| | 720 | **0.412** | **0.424** | 0.427 | 0.431 | 0.428 | 0.432 | 0.433 | 0.427 | - | - | - | - | 0.422 | 0.425 | 0.721 | 0.571 |
| | Avg | **0.348** | **0.385** | 0.356 | 0.387 | 0.366 | 0.391 | 0.373 | 0.389 | - | - | - | - | 0.356 | 0.385 | 0.699 | 0.557 |
| ETTm2 | 96 | **0.172** | 0.261 | 0.175 | 0.268 | 0.174 | **0.258** | 0.215 | 0.296 | - | - | - | - | 0.242 | 0.333 | 0.224 | 0.303 |
| | 192 | **0.223** | **0.300** | 0.240 | 0.312 | 0.257 | 0.303 | 0.259 | 0.323 | - | - | - | - | 0.283 | 0.345 | 0.273 | 0.331 |
| | 336 | 0.282 | **0.331** | 0.298 | 0.351 | **0.277** | 0.333 | 0.319 | 0.364 | - | - | - | - | 0.303 | 0.349 | 0.399 | 0.402 |
| | 720 | 0.374 | **0.388** | 0.403 | 0.413 | **0.360** | 0.404 | 0.395 | 0.405 | - | - | - | - | 0.431 | 0.431 | 0.406 | 0.408 |
| | Avg | **0.263** | **0.320** | 0.279 | 0.336 | 0.267 | 0.325 | 0.297 | 0.347 | - | - | - | - | 0.314 | 0.365 | 0.326 | 0.361 |
| Weather | 96 | 0.158 | 0.211 | 0.166 | 0.216 | **0.153** | **0.196** | 0.162 | 0.214 | - | - | - | - | 0.216 | 0.280 | 0.154 | 0.205 |
| | 192 | **0.199** | 0.249 | 0.208 | 0.254 | 0.214 | 0.253 | 0.203 | 0.252 | - | - | - | - | 0.303 | 0.335 | 0.200 | **0.243** |
| | 336 | 0.246 | 0.286 | 0.257 | 0.290 | **0.243** | **0.272** | 0.260 | 0.297 | - | - | - | - | 0.351 | 0.358 | 0.252 | 0.286 |
| | 720 | **0.317** | 0.337 | 0.326 | 0.338 | 0.324 | 0.349 | 0.330 | 0.342 | - | - | - | - | 0.425 | 0.343 | 0.324 | **0.335** |
| | Avg | **0.230** | 0.271 | 0.239 | 0.275 | 0.234 | **0.265** | 0.239 | 0.276 | - | - | - | - | 0.324 | 0.329 | 0.233 | 0.267 |
| Electricity | 96 | **0.133** | **0.223** | 0.190 | 0.279 | 0.163 | 0.255 | 0.186 | 0.268 | - | - | - | - | 0.197 | 0.277 | 0.195 | 0.275 |
| | 192 | **0.147** | **0.237** | 0.195 | 0.285 | 0.194 | 0.288 | 0.193 | 0.276 | - | - | - | - | 0.197 | 0.279 | 0.195 | 0.277 |
| | 336 | **0.166** | **0.265** | 0.211 | 0.301 | 0.201 | 0.298 | 0.206 | 0.289 | - | - | - | - | 0.211 | 0.295 | 0.210 | 0.294 |
| | 720 | **0.203** | **0.297** | 0.253 | 0.333 | 0.263 | 0.343 | 0.250 | 0.324 | - | - | - | - | 0.255 | 0.330 | 0.252 | 0.327 |
| | Avg | **0.162** | **0.256** | 0.212 | 0.300 | 0.205 | 0.296 | 0.209 | 0.289 | - | - | - | - | 0.215 | 0.295 | 0.213 | 0.293 |
| Traffic | 96 | **0.368** | **0.262** | 0.471 | 0.309 | 0.448 | 0.298 | 0.595 | 0.360 | - | - | - | - | 0.378 | 0.365 | 0.480 | 0.357 |
| | 192 | 0.373 | **0.251** | 0.475 | 0.308 | 0.445 | 0.301 | 0.576 | 0.353 | - | - | - | - | 0.371 | 0.352 | 0.439 | 0.336 |
| | 336 | **0.395** | **0.254** | 0.490 | 0.315 | 0.492 | 0.320 | 0.569 | 0.362 | - | - | - | - | 0.467 | 0.354 | 0.460 | 0.344 |
| | 720 | **0.432** | **0.290** | 0.524 | 0.332 | 0.514 | 0.321 | 0.603 | 0.372 | - | - | - | - | 0.525 | 0.378 | 0.499 | 0.364 |
| | Avg | **0.392** | **0.264** | 0.490 | 0.316 | 0.475 | 0.310 | 0.586 | 0.362 | - | - | - | - | 0.435 | 0.362 | 0.470 | 0.350 |

Table 17: Complete results of long-term forecasting tasks for the cross-domain setting of forecasting the future $O \in \{96, 192, 336, 720\}$ time points based on the past 336 time points. **All the results of baselines are based on the encoder utilized in their original papers.** The standard deviations of SimMTM are within 0.005 for MSE and within 0.004 for MAE.

| Models | | **SimMTM** | | Random init. | | Ti-MAE [21] | | TST [56] | | LaST [42] | | TF-C [57] | | CoST [46] | | TS2Vec [55] | |
|---|---|---|---|---|---|---|---|---|---|---|---|---|---|---|---|---|---|
| Metric | | MSE | MAE | MSE | MAE | MSE | MAE | MSE | MAE | MSE | MAE | MSE | MAE | MSE | MAE | MSE | MAE |
| ETTh2 → ETTh1 | 96 | 0.372 | **0.401** | 0.380 | 0.412 | 0.703 | 0.562 | 0.653 | 0.468 | **0.362** | 0.420 | 0.596 | 0.569 | 0.378 | 0.421 | 0.849 | 0.694 |
| | 192 | **0.414** | **0.425** | 0.416 | 0.434 | 0.715 | 0.567 | 0.658 | 0.502 | 0.426 | 0.478 | 0.614 | 0.621 | 0.424 | 0.451 | 0.909 | 0.738 |
| | 336 | **0.429** | **0.436** | 0.448 | 0.458 | 0.733 | 0.579 | 0.631 | 0.561 | 0.522 | 0.509 | 0.694 | 0.664 | 0.651 | 0.582 | 1.082 | 0.775 |
| | 720 | **0.446** | **0.458** | 0.481 | 0.487 | 0.762 | 0.622 | 0.638 | 0.608 | 0.460 | 0.478 | 0.635 | 0.683 | 0.883 | 0.701 | 0.934 | 0.769 |
| | Avg | **0.415** | **0.430** | 0.431 | 0.448 | 0.728 | 0.583 | 0.645 | 0.535 | 0.443 | 0.471 | 0.635 | 0.634 | 0.584 | 0.539 | 0.944 | 0.744 |
| ETTm1 → ETTh1 | 96 | 0.367 | 0.398 | 0.380 | 0.412 | 0.715 | 0.581 | 0.627 | 0.477 | **0.360** | **0.374** | 0.666 | 0.647 | 0.423 | 0.450 | 0.991 | 0.765 |
| | 192 | 0.396 | 0.421 | 0.416 | 0.434 | 0.729 | 0.587 | 0.628 | 0.500 | **0.381** | **0.371** | 0.672 | 0.653 | 0.641 | 0.578 | 0.829 | 0.699 |
| | 336 | 0.471 | **0.437** | **0.448** | 0.458 | 0.712 | 0.583 | 0.683 | 0.554 | 0.472 | 0.531 | 0.626 | 0.711 | 0.863 | 0.694 | 0.971 | 0.787 |
| | 720 | **0.454** | **0.463** | 0.481 | 0.487 | 0.747 | 0.627 | 0.642 | 0.600 | 0.490 | 0.488 | 0.835 | 0.797 | 1.071 | 0.805 | 1.037 | 0.820 |
| | Avg | **0.422** | **0.430** | 0.431 | 0.448 | 0.726 | 0.595 | 0.645 | 0.533 | 0.426 | 0.441 | 0.700 | 0.702 | 0.750 | 0.632 | 0.957 | 0.768 |
| ETTm2 → ETTh1 | 96 | 0.388 | 0.421 | 0.380 | **0.412** | 0.699 | 0.566 | 0.559 | 0.489 | 0.428 | 0.454 | 0.968 | 0.738 | **0.377** | 0.419 | 0.783 | 0.669 |
| | 192 | 0.419 | **0.423** | **0.416** | 0.434 | 0.722 | 0.573 | 0.600 | 0.579 | 0.427 | 0.497 | 1.080 | 0.801 | 0.422 | 0.450 | 0.828 | 0.691 |
| | 336 | **0.435** | **0.444** | 0.448 | 0.458 | 0.714 | 0.569 | 0.677 | 0.572 | 0.528 | 0.540 | 1.091 | 0.824 | 0.648 | 0.580 | 0.990 | 0.762 |
| | 720 | **0.468** | **0.474** | 0.481 | 0.487 | 0.760 | 0.611 | 0.694 | 0.664 | 0.527 | 0.537 | 1.226 | 0.893 | 0.880 | 0.699 | 0.985 | 0.783 |
| | Avg | **0.428** | **0.441** | 0.431 | 0.448 | 0.724 | 0.580 | 0.632 | 0.576 | 0.503 | 0.507 | 1.091 | 0.814 | 0.582 | 0.537 | 0.896 | 0.726 |
| Weather → ETTh1 | 96 | 0.477 | 0.444 | **0.380** | **0.412** | - | - | - | - | - | - | - | - | - | - | - | - |
| | 192 | 0.454 | 0.522 | **0.416** | **0.434** | - | - | - | - | - | - | - | - | - | - | - | - |
| | 336 | **0.424** | **0.434** | 0.448 | 0.458 | - | - | - | - | - | - | - | - | - | - | - | - |
| | 720 | **0.468** | **0.469** | 0.481 | 0.487 | - | - | - | - | - | - | - | - | - | - | - | - |
| | Avg | 0.456 | 0.467 | **0.431** | **0.448** | - | - | - | - | - | - | - | - | - | - | - | - |
| ETTh1 → ETTm1 | 96 | 0.290 | 0.348 | 0.295 | 0.346 | 0.667 | 0.521 | 0.425 | 0.381 | 0.295 | 0.387 | 0.672 | 0.600 | **0.248** | **0.332** | 0.605 | 0.561 |
| | 192 | **0.327** | **0.372** | 0.333 | 0.374 | 0.561 | 0.479 | 0.495 | 0.478 | 0.335 | 0.379 | 0.721 | 0.639 | 0.336 | 0.391 | 0.615 | 0.561 |
| | 336 | **0.357** | 0.392 | 0.370 | 0.398 | 0.690 | 0.533 | 0.456 | 0.441 | 0.379 | **0.363** | 0.755 | 0.664 | 0.381 | 0.421 | 0.763 | 0.677 |
| | 720 | 0.409 | **0.423** | 0.427 | 0.431 | 0.744 | 0.583 | 0.554 | 0.477 | **0.403** | 0.431 | 0.837 | 0.705 | 0.469 | 0.482 | 0.805 | 0.664 |
| | Avg | **0.346** | **0.384** | 0.356 | 0.387 | 0.666 | 0.529 | 0.482 | 0.444 | 0.353 | 0.390 | 0.746 | 0.652 | 0.359 | 0.407 | 0.697 | 0.616 |
| ETTh2 → ETTm1 | 96 | 0.322 | 0.347 | 0.295 | 0.346 | 0.658 | 0.505 | 0.449 | 0.343 | 0.314 | 0.396 | 0.677 | 0.603 | **0.253** | **0.342** | 0.466 | 0.480 |
| | 192 | **0.332** | **0.372** | 0.333 | 0.374 | 0.594 | 0.511 | 0.477 | 0.407 | 0.587 | 0.545 | 0.718 | 0.638 | 0.367 | 0.392 | 0.557 | 0.532 |
| | 336 | 0.394 | 0.391 | **0.370** | **0.398** | 0.732 | 0.532 | 0.407 | 0.519 | 0.631 | 0.584 | 0.755 | 0.663 | 0.388 | 0.431 | 0.646 | 0.576 |
| | 720 | **0.411** | **0.424** | 0.427 | 0.431 | 0.768 | 0.592 | 0.557 | 0.523 | 0.368 | 0.429 | 0.848 | 0.712 | 0.498 | 0.488 | 0.752 | 0.638 |
| | Avg | 0.365 | 0.384 | **0.356** | **0.387** | 0.688 | 0.535 | 0.472 | 0.448 | 0.475 | 0.489 | 0.750 | 0.654 | 0.377 | 0.413 | 0.606 | 0.556 |
| ETTm2 → ETTm1 | 96 | 0.297 | 0.348 | 0.295 | 0.346 | 0.647 | 0.497 | 0.471 | 0.422 | 0.304 | 0.388 | 0.610 | 0.577 | **0.239** | **0.331** | 0.586 | 0.515 |
| | 192 | **0.332** | **0.370** | 0.333 | 0.374 | 0.597 | 0.508 | 0.495 | 0.442 | 0.429 | 0.494 | 0.725 | 0.657 | 0.339 | 0.371 | 0.624 | 0.562 |
| | 336 | **0.364** | **0.393** | 0.370 | 0.398 | 0.700 | 0.525 | 0.455 | 0.424 | 0.499 | 0.523 | 0.768 | 0.684 | 0.371 | 0.421 | 1.035 | 0.806 |
| | 720 | **0.410** | **0.421** | 0.427 | 0.431 | 0.786 | 0.596 | 0.498 | 0.532 | 0.422 | 0.450 | 0.927 | 0.759 | 0.467 | 0.481 | 0.780 | 0.669 |
| | Avg | **0.351** | **0.383** | 0.356 | 0.387 | 0.682 | 0.531 | 0.480 | 0.455 | 0.414 | 0.464 | 0.758 | 0.669 | 0.354 | 0.401 | 0.756 | 0.638 |
| Weather → ETTm1 | 96 | 0.304 | 0.354 | **0.295** | **0.346** | - | - | - | - | - | - | - | - | - | - | - | - |
| | 192 | 0.338 | 0.375 | **0.333** | **0.374** | - | - | - | - | - | - | - | - | - | - | - | - |
| | 336 | 0.371 | **0.397** | **0.370** | 0.398 | - | - | - | - | - | - | - | - | - | - | - | - |
| | 720 | **0.417** | **0.426** | 0.427 | 0.431 | - | - | - | - | - | - | - | - | - | - | - | - |
| | Avg | 0.358 | 0.388 | **0.356** | **0.387** | - | - | - | - | - | - | - | - | - | - | - | - |

Table 18: Complete results of long-term forecasting tasks for the in-domain setting. **All the results of baseline are based on the unified channel-independent Transformer encoder.** The past sequence length is set as 336. The unified channel-independent transformer model can perform the transfer experiment between datasets with different variables. The standard deviations of SimMTM are within 0.005 for MSE and within 0.004 for MAE.

| Models | | SimMTM | | Random init. | | Ti-MAE [21] | | TST [56] | | LaST [42] | | TF-C [57] | | CoST [46] | | TS2Vec [55] |
|---|---|---|---|---|---|---|---|---|---|---|---|---|---|---|---|---|---|
| Metric | | MSE | MAE | MSE | MAE | MSE | MAE | MSE | MAE | MSE | MAE | MSE | MAE | MSE | MAE | MSE | MAE |
| ETTh2 ↓ ETTh1 | 96 | **0.372** | 0.401 | 0.380 | 0.412 | 0.399 | 0.424 | 0.401 | 0.425 | - | - | - | - | 0.376 | **0.362** | 0.436 | 0.430 |
| | 192 | 0.414 | 0.425 | 0.416 | 0.434 | 0.454 | 0.440 | 0.531 | 0.484 | - | - | - | - | **0.376** | **0.362** | 0.455 | 0.440 |
| | 336 | **0.429** | **0.436** | 0.448 | 0.458 | 0.497 | 0.469 | 0.474 | 0.459 | - | - | - | - | 0.444 | 0.444 | 0.689 | 0.584 |
| | 720 | **0.446** | 0.458 | 0.481 | 0.487 | 0.515 | 0.492 | 0.471 | 0.469 | - | - | - | - | 0.517 | 0.510 | 0.489 | 0.490 |
| | Avg | **0.415** | **0.430** | 0.431 | 0.448 | 0.466 | 0.456 | 0.469 | 0.459 | - | - | - | - | 0.428 | 0.433 | 0.517 | 0.486 |
| ETTm1 ↓ ETTh1 | 96 | **0.367** | **0.398** | 0.380 | 0.412 | 0.400 | 0.418 | 0.443 | 0.440 | - | - | - | - | 0.465 | 0.456 | 0.413 | 0.443 |
| | 192 | **0.396** | **0.421** | 0.416 | 0.434 | 0.434 | 0.445 | 0.471 | 0.455 | - | - | - | - | 0.722 | 0.588 | 0.459 | 0.465 |
| | 336 | 0.471 | **0.437** | **0.448** | 0.458 | 0.510 | 0.467 | 0.462 | 0.455 | - | - | - | - | 0.712 | 0.586 | 0.614 | 0.554 |
| | 720 | 0.454 | **0.463** | 0.481 | 0.487 | 0.636 | 0.544 | 0.525 | 0.503 | - | - | - | - | 0.581 | 0.533 | **0.450** | 0.464 |
| | Avg | **0.422** | **0.430** | 0.431 | 0.448 | 0.495 | 0.469 | 0.475 | 0.463 | - | - | - | - | 0.620 | 0.541 | 0.484 | 0.482 |
| ETTm2 ↓ ETTh1 | 96 | 0.388 | 0.421 | **0.380** | **0.412** | 0.433 | 0.431 | 0.389 | 0.413 | - | - | - | - | 0.403 | 0.426 | 0.483 | 0.480 |
| | 192 | 0.419 | **0.423** | **0.416** | 0.434 | 0.474 | 0.458 | 0.463 | 0.452 | - | - | - | - | 0.457 | 0.468 | 0.579 | 0.537 |
| | 336 | **0.435** | 0.444 | 0.448 | 0.458 | 0.515 | 0.448 | 0.492 | 0.465 | - | - | - | - | 0.794 | 0.682 | 0.673 | 0.563 |
| | 720 | **0.468** | **0.474** | 0.481 | 0.487 | 0.496 | 0.488 | 0.468 | 0.468 | - | - | - | - | 0.739 | 0.617 | 0.729 | 0.620 |
| | Avg | **0.428** | **0.441** | 0.431 | 0.448 | 0.464 | 0.456 | 0.453 | 0.450 | - | - | - | - | 0.598 | 0.548 | 0.616 | 0.550 |
| Weather ↓ ETTh1 | 96 | 0.477 | 0.444 | **0.380** | 0.412 | 0.397 | 0.440 | 0.428 | 0.429 | - | - | - | - | 0.421 | 0.410 | 0.393 | **0.410** |
| | 192 | 0.454 | 0.522 | **0.416** | **0.434** | 0.458 | 0.466 | 0.461 | 0.451 | - | - | - | - | 0.539 | 0.503 | 0.440 | 0.437 |
| | 336 | **0.424** | **0.434** | 0.448 | 0.458 | 0.479 | 0.458 | 0.463 | 0.456 | - | - | - | - | 0.568 | 0.514 | 0.450 | 0.451 |
| | 720 | **0.468** | **0.469** | 0.481 | 0.487 | 0.515 | 0.492 | 0.507 | 0.489 | - | - | - | - | 0.544 | 0.522 | 0.567 | 0.541 |
| | Avg | 0.456 | 0.467 | **0.431** | **0.448** | 0.462 | 0.464 | 0.465 | 0.456 | - | - | - | - | 0.518 | 0.487 | 0.463 | 0.460 |
| ETTh1 ↓ ETTm1 | 96 | **0.290** | 0.348 | 0.295 | **0.346** | 0.311 | 0.355 | 0.315 | 0.354 | - | - | - | - | 0.308 | 0.355 | 0.681 | 0.545 |
| | 192 | 0.327 | **0.372** | 0.333 | 0.374 | 0.337 | **0.372** | 0.365 | 0.391 | - | - | - | - | 0.357 | 0.390 | 0.689 | 0.551 |
| | 336 | **0.357** | **0.392** | 0.370 | 0.398 | 0.372 | 0.398 | 0.384 | 0.400 | - | - | - | - | 0.396 | 0.402 | 0.705 | 0.560 |
| | 720 | **0.409** | **0.423** | 0.427 | 0.431 | 0.422 | 0.433 | 0.428 | 0.426 | - | - | - | - | 0.419 | 0.423 | 0.722 | 0.571 |
| | Avg | **0.346** | **0.384** | 0.356 | 0.387 | 0.360 | 0.390 | 0.373 | 0.393 | - | - | - | - | 0.370 | 0.393 | 0.699 | 0.557 |
| ETTh2 ↓ ETTm1 | 96 | 0.322 | 0.347 | **0.295** | **0.346** | 0.323 | 0.362 | 0.338 | 0.383 | - | - | - | - | 0.322 | 0.351 | 0.679 | 0.546 |
| | 192 | 0.332 | **0.372** | 0.333 | 0.374 | 0.370 | 0.395 | 0.394 | 0.408 | - | - | - | - | **0.331** | 0.373 | 0.673 | 0.551 |
| | 336 | 0.394 | **0.391** | **0.370** | 0.398 | 0.397 | 0.413 | 0.401 | 0.412 | - | - | - | - | 0.382 | 0.397 | 0.703 | 0.557 |
| | 720 | **0.411** | **0.424** | 0.427 | 0.431 | 0.442 | 0.439 | 0.434 | 0.432 | - | - | - | - | 0.417 | 0.428 | 0.722 | 0.573 |
| | Avg | 0.365 | **0.384** | **0.356** | 0.387 | 0.383 | 0.402 | 0.391 | 0.409 | - | - | - | - | 0.363 | 0.387 | 0.694 | 0.557 |
| ETTm2 ↓ ETTm1 | 96 | 0.297 | 0.348 | **0.295** | **0.346** | 0.333 | 0.378 | 0.327 | 0.364 | - | - | - | - | 0.320 | 0.364 | 0.422 | 0.434 |
| | 192 | **0.332** | **0.370** | 0.333 | 0.374 | 0.381 | 0.398 | 0.362 | 0.389 | - | - | - | - | 0.367 | 0.386 | 0.387 | 0.371 |
| | 336 | **0.364** | **0.393** | 0.370 | 0.398 | 0.394 | 0.413 | 0.401 | 0.418 | - | - | - | - | 0.374 | 0.394 | 0.402 | 0.444 |
| | 720 | **0.410** | **0.421** | 0.427 | 0.431 | 0.455 | 0.453 | 0.437 | 0.437 | - | - | - | - | 0.479 | 0.503 | 0.481 | 0.432 |
| | Avg | **0.351** | **0.383** | 0.356 | 0.387 | 0.390 | 0.410 | 0.382 | 0.402 | - | - | - | - | 0.385 | 0.412 | 0.423 | 0.420 |
| Weather ↓ ETTm1 | 96 | **0.294** | 0.354 | 0.295 | **0.346** | 0.338 | 0.380 | 0.324 | 0.366 | - | - | - | - | 0.324 | 0.360 | 0.329 | 0.359 |
| | 192 | **0.318** | **0.355** | 0.333 | 0.374 | 0.473 | 0.457 | 0.349 | 0.377 | - | - | - | - | 0.359 | 0.387 | 0.392 | 0.392 |
| | 336 | **0.361** | **0.397** | 0.370 | 0.398 | 0.402 | 0.415 | 0.378 | 0.398 | - | - | - | - | 0.395 | 0.399 | 0.372 | 0.400 |
| | 720 | 0.427 | **0.426** | 0.427 | 0.431 | 0.432 | 0.438 | **0.422** | 0.427 | - | - | - | - | 0.450 | 0.467 | 0.434 | 0.429 |
| | Avg | **0.350** | **0.383** | 0.356 | 0.387 | 0.411 | 0.423 | 0.368 | 0.392 | - | - | - | - | 0.382 | 0.403 | 0.382 | 0.395 |

Table 19: Full ablation studies for the in-domain setting of forecasting. The standard deviations of SimMTM are within 0.005 for MSE and within 0.004 for MAE.

| Input-336 | | Supervised | | W/o $\mathcal{L}_{\text{reconstruction}}$ | | W/o $\mathcal{L}_{\text{constraint}}$ | | **SimMTM** | |
|---|---|---|---|---|---|---|---|---|---|
| Metric | | MSE | MAE | MSE | MAE | MSE | MAE | MSE | MAE |
| ETTh1 | 96 | 0.380 | 0.412 | **0.377** | 0.408 | 0.381 | 0.409 | 0.379 | **0.407** |
| | 192 | 0.416 | 0.434 | 0.419 | 0.443 | **0.409** | 0.443 | 0.412 | **0.424** |
| | 336 | 0.448 | 0.458 | 0.423 | 0.434 | 0.432 | 0.444 | **0.421** | **0.431** |
| | 720 | 0.481 | 0.487 | 0.437 | 0.454 | 0.447 | 0.454 | **0.424** | **0.449** |
| | Avg | 0.431 | 0.448 | 0.414 | 0.435 | 0.417 | 0.438 | **0.409** | **0.428** |
| ETTh2 | 96 | 0.325 | 0.374 | **0.288** | **0.344** | 0.312 | 0.365 | 0.293 | 0.347 |
| | 192 | 0.400 | 0.424 | 0.356 | 0.391 | 0.389 | 0.418 | **0.355** | **0.386** |
| | 336 | 0.405 | 0.433 | **0.368** | 0.406 | 0.396 | 0.432 | 0.370 | **0.401** |
| | 720 | 0.451 | 0.475 | 0.409 | 0.432 | 0.448 | 0.479 | **0.395** | **0.427** |
| | Avg | 0.395 | 0.427 | 0.355 | 0.393 | 0.386 | 0.424 | **0.353** | **0.390** |
| ETTm1 | 96 | 0.295 | 0.346 | 0.291 | 0.343 | **0.282** | **0.337** | 0.288 | 0.348 |
| | 192 | 0.333 | 0.374 | 0.330 | 0.390 | 0.324 | 0.388 | **0.327** | **0.373** |
| | 336 | 0.370 | 0.398 | 0.369 | 0.399 | 0.366 | 0.397 | **0.363** | **0.395** |
| | 720 | 0.427 | 0.431 | 0.417 | 0.429 | 0.424 | 0.435 | **0.412** | **0.424** |
| | Avg | 0.356 | 0.387 | 0.352 | 0.390 | 0.349 | 0.389 | **0.348** | **0.385** |
| ETTm2 | 96 | 0.175 | 0.268 | 0.174 | 0.265 | **0.170** | **0.261** | 0.172 | **0.261** |
| | 192 | 0.240 | 0.312 | 0.232 | 0.303 | 0.244 | 0.320 | **0.223** | **0.300** |
| | 336 | 0.298 | 0.351 | 0.313 | 0.365 | **0.279** | 0.334 | 0.282 | **0.331** |
| | 720 | 0.403 | 0.413 | 0.376 | 0.451 | 0.376 | **0.378** | 0.374 | 0.388 |
| | Avg | 0.279 | 0.336 | 0.274 | 0.346 | 0.267 | 0.323 | **0.263** | **0.320** |
| Weather | 96 | 0.166 | 0.216 | 0.164 | **0.209** | 0.160 | 0.212 | **0.158** | 0.211 |
| | 192 | 0.208 | 0.254 | 0.203 | 0.258 | 0.203 | 0.251 | **0.199** | **0.249** |
| | 336 | 0.257 | 0.290 | 0.244 | 0.289 | 0.253 | 0.290 | **0.246** | **0.286** |
| | 720 | 0.326 | 0.338 | 0.322 | 0.343 | 0.325 | 0.340 | **0.317** | **0.337** |
| | Avg | 0.239 | 0.275 | 0.233 | 0.275 | 0.235 | 0.273 | **0.230** | **0.271** |
| Electricity | 96 | 0.190 | 0.279 | 0.177 | 0.270 | 0.134 | **0.220** | **0.133** | 0.223 |
| | 192 | 0.195 | 0.285 | 0.184 | 0.279 | 0.163 | 0.274 | **0.147** | **0.237** |
| | 336 | 0.211 | 0.301 | 0.202 | 0.300 | 0.223 | 0.311 | **0.166** | **0.265** |
| | 720 | 0.253 | 0.333 | 0.250 | 0.337 | 0.241 | 0.321 | **0.203** | **0.297** |
| | Avg | 0.212 | 0.300 | 0.203 | 0.397 | 0.190 | 0.282 | **0.162** | **0.256** |
| Traffic | 96 | 0.471 | 0.309 | **0.366** | **0.257** | 0.457 | 0.301 | 0.368 | 0.262 |
| | 192 | 0.475 | 0.308 | **0.373** | 0.266 | 0.468 | 0.325 | **0.373** | **0.251** |
| | 336 | 0.490 | 0.315 | 0.401 | 0.249 | 0.487 | 0.302 | **0.395** | **0.254** |
| | 720 | 0.524 | 0.332 | 0.472 | 0.312 | 0.485 | 0.315 | **0.432** | **0.290** |
| | Avg | 0.490 | 0.316 | 0.403 | 0.271 | 0.474 | 0.311 | **0.392** | **0.264** |

Table 20: Full ablation studies on transfer to ETTh1 and ETTm1 for the cross-domain setting of forecasting. The standard deviations of SimMTM are within 0.005 for MSE and within 0.004 for MAE.

| Input-336 | | Supervised | | W/o $\mathcal{L}_{\text{reconstruction}}$ | | W/o $\mathcal{L}_{\text{constraint}}$ | | **SimMTM** | |
|---|---|---|---|---|---|---|---|---|---|
| Metric | | MSE | MAE | MSE | MAE | MSE | MAE | MSE | MAE |
| ETTh2 ↓ ETTh1 | 96 | 0.380 | 0.412 | 0.377 | **0.400** | 0.402 | 0.411 | **0.372** | 0.401 |
| | 192 | 0.416 | 0.434 | 0.417 | 0.424 | 0.417 | **0.420** | **0.414** | 0.425 |
| | 336 | 0.448 | 0.458 | 0.437 | 0.439 | 0.437 | **0.435** | **0.429** | 0.436 |
| | 720 | 0.481 | 0.487 | 0.448 | 0.463 | 0.456 | 0.467 | **0.446** | **0.458** |
| | Avg | 0.431 | 0.448 | 0.420 | 0.432 | 0.423 | **0.430** | **0.415** | 0.430 |
| ETTm1 ↓ ETTh1 | 96 | 0.380 | 0.412 | 0.382 | **0.397** | 0.375 | 0.399 | **0.367** | 0.398 |
| | 192 | 0.416 | 0.434 | 0.418 | **0.418** | 0.413 | 0.422 | **0.396** | 0.421 |
| | 336 | 0.448 | 0.458 | 0.437 | **0.434** | **0.434** | 0.438 | 0.471 | 0.437 |
| | 720 | 0.481 | 0.487 | 0.459 | 0.469 | 0.467 | 0.475 | **0.454** | **0.463** |
| | Avg | 0.431 | 0.448 | 0.424 | 0.430 | 0.422 | 0.434 | **0.422** | **0.430** |
| ETTm2 ↓ ETTh1 | 96 | **0.380** | **0.412** | 0.388 | 0.418 | 0.384 | 0.415 | 0.388 | 0.421 |
| | 192 | **0.416** | 0.434 | 0.429 | 0.444 | 0.423 | 0.439 | 0.419 | **0.423** |
| | 336 | 0.448 | 0.458 | 0.467 | 0.472 | 0.458 | 0.465 | **0.435** | **0.444** |
| | 720 | 0.481 | 0.487 | 0.521 | 0.507 | 0.501 | 0.497 | **0.468** | **0.474** |
| | Avg | 0.431 | 0.448 | 0.451 | 0.460 | 0.441 | 0.454 | **0.428** | **0.441** |
| Weather ↓ ETTh1 | 96 | **0.380** | 0.412 | 0.385 | **0.400** | 0.394 | 0.406 | 0.477 | 0.444 |
| | 192 | **0.416** | 0.434 | 0.417 | 0.429 | 0.425 | **0.424** | 0.454 | 0.522 |
| | 336 | 0.448 | 0.458 | 0.434 | **0.434** | 0.441 | 0.439 | **0.424** | **0.434** |
| | 720 | 0.481 | 0.487 | **0.444** | **0.464** | 0.446 | 0.468 | 0.468 | 0.469 |
| | Avg | 0.431 | 0.448 | **0.420** | **0.432** | 0.427 | 0.434 | 0.456 | 0.467 |
| ETTh1 ↓ ETTm1 | 96 | 0.295 | 0.346 | **0.286** | **0.341** | 0.290 | 0.346 | 0.290 | 0.348 |
| | 192 | 0.333 | 0.374 | **0.322** | **0.362** | 0.353 | 0.388 | 0.327 | 0.372 |
| | 336 | 0.370 | 0.398 | 0.362 | 0.418 | 0.362 | 0.412 | **0.357** | **0.392** |
| | 720 | 0.427 | 0.431 | 0.417 | 0.431 | 0.422 | 0.432 | **0.409** | **0.423** |
| | Avg | 0.356 | 0.387 | 0.347 | 0.388 | 0.357 | 0.395 | **0.346** | **0.384** |
| ETTh2 ↓ ETTm1 | 96 | **0.295** | **0.346** | 0.299 | 0.348 | 0.301 | 0.352 | 0.322 | 0.347 |
| | 192 | 0.333 | 0.374 | **0.324** | 0.366 | 0.332 | **0.359** | 0.332 | 0.372 |
| | 336 | **0.370** | 0.398 | 0.374 | 0.401 | 0.389 | **0.382** | 0.394 | 0.391 |
| | 720 | 0.427 | 0.431 | 0.415 | **0.419** | 0.421 | 0.442 | **0.411** | 0.424 |
| | Avg | 0.356 | 0.387 | **0.353** | 0.386 | 0.361 | **0.384** | 0.365 | **0.384** |
| ETTm2 ↓ ETTm1 | 96 | 0.295 | 0.346 | 0.299 | 0.351 | **0.285** | **0.336** | 0.297 | 0.348 |
| | 192 | 0.333 | 0.374 | 0.334 | 0.372 | 0.343 | **0.366** | **0.332** | 0.370 |
| | 336 | 0.370 | 0.398 | 0.362 | **0.388** | 0.360 | 0.399 | 0.364 | 0.393 |
| | 720 | 0.427 | 0.431 | 0.417 | 0.431 | 0.422 | 0.432 | **0.410** | **0.421** |
| | Avg | 0.356 | 0.387 | 0.353 | 0.386 | 0.353 | **0.383** | **0.351** | 0.383 |
| Weather ↓ ETTm1 | 96 | 0.295 | **0.346** | 0.322 | 0.361 | 0.309 | 0.354 | **0.294** | 0.354 |
| | 192 | 0.333 | 0.374 | 0.344 | 0.378 | 0.343 | 0.365 | **0.318** | **0.355** |
| | 336 | 0.370 | 0.398 | 0.371 | 0.399 | 0.401 | 0.411 | **0.361** | **0.397** |
| | 720 | 0.427 | 0.431 | 0.426 | **0.422** | **0.425** | 0.427 | 0.427 | 0.426 |
| | Avg | 0.356 | 0.387 | 0.366 | 0.390 | 0.370 | 0.389 | **0.350** | **0.383** |

Table 21: Full results for applying SimMTM to four advanced time series forecasting models under the in-domain setting. The gray mark represents negative transfer (↓).

| Models | Transformer [39] | | | | Autoformer [47] | | | | Ns Transformer [24] | | | | PatchTST [26] | | | | | |
|---|---|---|---|---|---|---|---|---|---|---|---|---|---|---|---|---|---|---|
| | Random init. | | +SimMTM | | Random init. | | +SimMTM | | Random init. | | +SimMTM | | Random init. | | +Sub-serie Masking | | +SimMTM | |
| Metric | MSE | MAE | MSE | MAE | MSE | MAE | MSE | MAE | MSE | MAE | MSE | MAE | MSE | MAE | MSE | MAE | MSE | MAE |
| ETTh1 96 | 0.847 | 0.731 | 0.775 | 0.691 | 0.536 | 0.548 | 0.526 | 0.536 | 0.513 | 0.491 | 0.490 | 0.489 | 0.375 | 0.399 | 0.366 | 0.398 | 0.373 | 0.399 |
| ETTh1 192 | 1.084 | 0.841 | 0.918 | 0.763 | 0.543 | 0.551 | 0.523 | 0.548 | 0.534 | 0.504 | 0.517 | 0.499 | 0.414 | 0.421 | 0.431 | 0.443 | 0.406 | 0.428 |
| ETTh1 336 | 1.350 | 0.956 | 1.079 | 0.845 | 0.615 | 0.592 | 0.595 | 0.591 | 0.588 | 0.535 | 0.552 | 0.520 | 0.431 | 0.436 | 0.450↓ | 0.456↓ | 0.422 | 0.431 |
| ETTh1 720 | 1.069 | 0.817 | 0.935 | 0.761 | 0.599 | 0.600 | 0.600 | 0.597 | 0.643 | 0.616 | 0.614 | 0.598 | 0.449 | 0.466 | 0.472↓ | 0.484↓ | 0.436 | 0.452 |
| ETTh1 Avg | 1.088 | 0.836 | **0.927** | **0.761** | 0.573 | 0.573 | **0.561** | **0.568** | 0.570 | 0.537 | **0.543** | **0.527** | 0.417 | 0.431 | 0.430↓ | 0.445↓ | **0.409** | **0.428** |
| ETTh2 96 | 2.029 | 1.150 | 1.879 | 1.104 | 0.492 | 0.517 | 0.488 | 0.514 | 0.476 | 0.458 | 0.445 | 0.448 | 0.274 | 0.336 | 0.284↓ | 0.343↓ | 0.274 | 0.337 |
| ETTh2 192 | 6.785 | 2.099 | 5.054 | 1.771 | 0.556 | 0.551 | 0.547 | 0.549 | 0.512 | 0.493 | 0.482 | 0.502 | 0.339 | 0.379 | 0.355↓ | 0.387↓ | 0.339 | 0.377 |
| ETTh2 336 | 4.568 | 1.711 | 4.242 | 1.658 | 0.572 | 0.578 | 0.563 | 0.570 | 0.552 | 0.551 | 0.512 | 0.537 | 0.331 | 0.380 | 0.379↓ | 0.411↓ | 0.327 | 0.381 |
| ETTh2 720 | 3.030 | 1.486 | 2.815 | 1.413 | 0.580 | 0.588 | 0.575 | 0.588 | 0.562 | 0.560 | 0.531 | 0.568 | 0.379 | 0.422 | 0.400↓ | 0.435↓ | 0.375 | 0.423 |
| ETTh2 Avg | 4.103 | 1.612 | **3.498** | **1.487** | 0.550 | 0.559 | **0.543** | **0.555** | 0.526 | 0.516 | **0.493** | **0.514** | 0.331 | 0.379 | 0.355↓ | 0.394↓ | **0.329** | 0.379 |
| ETTm1 96 | 0.562 | 0.520 | 0.513 | 0.497 | 0.523 | 0.488 | 0.482 | 0.465 | 0.386 | 0.398 | 0.340 | 0.376 | 0.290 | 0.342 | 0.289↓ | 0.344↓ | 0.288 | 0.343 |
| ETTm1 192 | 0.810 | 0.668 | 0.686 | 0.606 | 0.543 | 0.498 | 0.499 | 0.476 | 0.459 | 0.444 | 0.423 | 0.445 | 0.332 | 0.369 | 0.323 | 0.368 | 0.329 | 0.367 |
| ETTm1 336 | 1.096 | 0.814 | 1.003 | 0.760 | 0.675 | 0.551 | 0.601 | 0.524 | 0.495 | 0.464 | 0.423 | 0.459 | 0.366 | 0.392 | 0.353 | 0.387 | 0.361 | 0.387 |
| ETTm1 720 | 1.136 | 0.813 | 1.032 | 0.790 | 0.720 | 0.528 | 0.629 | 0.555 | 0.585 | 0.516 | 0.539 | 0.499 | 0.420 | 0.424 | 0.398 | 0.416 | 0.413 | 0.417 |
| ETTm1 Avg | 0.901 | 0.704 | **0.809** | **0.663** | 0.615 | 0.528 | **0.553** | **0.505** | 0.481 | 0.456 | **0.431** | **0.445** | 0.352 | 0.382 | **0.341** | 0.379 | 0.348 | **0.378** |
| ETTm2 96 | 0.508 | 0.539 | 0.336 | 0.425 | 0.255 | 0.339 | 0.255 | 0.340 | 0.192 | 0.274 | 0.188 | 0.277 | 0.165 | 0.255 | 0.166↓ | 0.256↓ | 0.163 | 0.253 |
| ETTm2 192 | 0.972 | 0.721 | 0.713 | 0.610 | 0.281 | 0.340 | 0.276 | 0.332 | 0.280 | 0.339 | 0.277 | 0.336 | 0.220 | 0.292 | 0.221↓ | 0.295↓ | 0.219 | 0.292 |
| ETTm2 336 | 1.419 | 0.897 | 1.517 | 0.942 | 0.339 | 0.372 | 0.309 | 0.359 | 0.334 | 0.361 | 0.325 | 0.355 | 0.278 | 0.329 | 0.278 | 0.333↓ | 0.275 | 0.328 |
| ETTm2 720 | 3.598 | 1.445 | 2.720 | 1.254 | 0.422 | 0.419 | 0.420 | 0.410 | 0.417 | 0.413 | 0.414 | 0.412 | 0.367 | 0.385 | 0.365 | 0.388↓ | 0.359 | 0.381 |
| ETTm2 Avg | 1.624 | 0.901 | **1.322** | **0.808** | 0.324 | 0.368 | **0.315** | **0.360** | 0.306 | 0.347 | **0.301** | **0.345** | 0.258 | 0.317 | 0.258 | 0.318↓ | **0.254** | **0.313** |

Table 22: Full results for fine-tuning to limited data scenarios. We fine-tune the model pre-trained from ETTh2 to ETTh1 with different data proportions {10%, 25%, 50%, 75%, 100%}.

| Models | **SimMTM** | | Random init. | | Ti-MAE[21] | | TST[56] | | LaST[42] | | TF-C[57] | | CoST[46] | | TS2Vec[55] | |
|---|---|---|---|---|---|---|---|---|---|---|---|---|---|---|---|---|
| Metric | MSE | MAE | MSE | MAE | MSE | MAE | MSE | MAE | MSE | MAE | MSE | MAE | MSE | MAE | MAE | MSE |
| ETTh2 → ETTh1 10% | **0.591** | **0.523** | 0.653 | 0.558 | 0.660 | 0.517 | 0.783 | 0.588 | 0.645 | 0.507 | 0.799 | 0.783 | 0.784 | 0.604 | 0.655 | 0.550 |
| 25% | **0.535** | **0.490** | 0.632 | 0.502 | 0.594 | 0.518 | 0.641 | 0.578 | 0.610 | 0.611 | 0.736 | 0.725 | 0.624 | 0.539 | 0.632 | 0.543 |
| 50% | **0.491** | **0.473** | 0.512 | 0.479 | 0.550 | 0.504 | 0.525 | 0.509 | 0.540 | 0.513 | 0.731 | 0.704 | 0.540 | 0.499 | 0.599 | 0.526 |
| 75% | **0.466** | **0.458** | 0.499 | 0.488 | 0.475 | 0.465 | 0.516 | 0.488 | 0.479 | 0.470 | 0.697 | 0.689 | 0.494 | 0.475 | 0.577 | 0.534 |
| 100% | **0.415** | **0.430** | 0.431 | 0.448 | 0.466 | 0.456 | 0.469 | 0.459 | 0.443 | 0.471 | 0.635 | 0.634 | 0.428 | 0.433 | 0.517 | 0.486 |

Table 23: In- and cross-domain settings of classification, where **all the baselines are based on the encoder utilized in their original papers**. For in-domain setting, we pre-train and fine-tune on the same dataset: Epilepsy. For cross-domain setting, we pre-train the model on SleepEEG and then fine-tune it on different datasets: Epilepsy, FD-B, Gesture, and EMG.

| | Scenarios | Models | Accuracy (%) | Precision (%) | Recall (%) | F1 (%) | Avg (%) |
|---|---|---|---|---|---|---|---|
| In-Domain | | Random init. | 89.83 | 92.13 | 74.47 | 79.59 | 84.00 |
| | Epilepsy ↓ Epilepsy | TS2vec [55] | 92.17 | 93.84 | 81.19 | 85.71 | 88.23 |
| | | CoST[46] | 88.07 | 91.58 | 66.05 | 69.11 | 78.70 |
| | | LaST [42] | 92.11 | 93.12 | 81.47 | 85.74 | 88.11 |
| | | TST [56] | 80.21 | 40.11 | 50.00 | 44.51 | 53.71 |
| | | Ti-MAE [21] | 90.09 | 93.90 | 77.24 | 78.21 | 84.86 |
| | | TF-C [57] | 93.96 | 94.87 | 85.82 | 89.46 | 91.03 |
| | | **SimMTM** | **94.75** | **95.60** | **89.93** | **91.41** | **92.92** |
| Cross-Domain | | Random init. | 89.83 | 92.13 | 74.47 | 79.59 | 84.00 |
| | SleepEEG ↓ Epilepsy | TS2vec [55] | 93.95 | 90.59 | 90.39 | 90.45 | 91.35 |
| | | CoST[46] | 88.40 | 88.20 | 72.34 | 76.88 | 81.45 |
| | | LaST [42] | 86.46 | 90.77 | 66.35 | 70.67 | 78.56 |
| | | TST [56] | 80.21 | 40.11 | 50.00 | 44.51 | 53.71 |
| | | Ti-MAE [21] | 89.71 | 72.36 | 67.47 | 68.55 | 74.52 |
| | | TF-C [57] | 94.95 | **94.56** | 89.08 | 91.49 | 92.52 |
| | | **SimMTM** | **95.49** | 93.36 | **92.28** | **92.81** | **93.49** |
| | | Random init. | 47.36 | 48.29 | 52.35 | 49.11 | 49.28 |
| | SleepEEG ↓ FD-B | TS2vec [55] | 47.90 | 43.39 | 48.42 | 43.89 | 45.90 |
| | | CoST[46] | 47.06 | 38.79 | 38.42 | 34.79 | 39.76 |
| | | LaST [42] | 46.67 | 43.90 | 47.71 | 45.17 | 45.86 |
| | | TST [56] | 46.40 | 41.58 | 45.50 | 41.34 | 43.71 |
| | | Ti-MAE [21] | 60.88 | 66.98 | 68.94 | 66.56 | 65.84 |
| | | TF-C [57] | 69.38 | **75.59** | 72.02 | 74.87 | 72.97 |
| | | **SimMTM** | **69.40** | 74.18 | **76.41** | **75.11** | **73.78** |
| | | Random init. | 42.19 | 47.51 | 49.63 | 48.86 | 47.05 |
| | SleepEEG ↓ Gesture | TS2vec [55] | 69.17 | 65.45 | 68.54 | 65.70 | 67.22 |
| | | CoST[46] | 68.33 | 65.30 | 68.33 | 66.42 | 67.09 |
| | | LaST [42] | 64.17 | 70.36 | 64.17 | 58.76 | 64.37 |
| | | TST [56] | 69.17 | 66.60 | 69.17 | 66.01 | 67.74 |
| | | Ti-MAE [21] | 71.88 | 70.35 | 76.75 | 68.37 | 71.84 |
| | | TF-C [57] | 76.42 | 77.31 | 74.29 | 75.72 | 75.94 |
| | | **SimMTM** | **80.00** | **79.03** | **80.00** | **78.67** | **79.43** |
| | | Random init. | 77.80 | 59.09 | 66.67 | 62.38 | 66.49 |
| | SleepEEG ↓ EMG | TS2vec [55] | 78.54 | 80.40 | 67.85 | 67.66 | 73.61 |
| | | CoST[46] | 53.65 | 49.07 | 42.10 | 35.27 | 45.02 |
| | | LaST [42] | 66.34 | 79.34 | 63.33 | 72.55 | 70.39 |
| | | TST [56] | 78.34 | 77.11 | 80.30 | 68.89 | 76.16 |
| | | Ti-MAE [21] | 69.99 | 70.25 | 63.44 | 70.89 | 68.64 |
| | | TF-C [57] | 81.71 | 72.65 | 81.59 | 76.83 | 78.20 |
| | | **SimMTM** | **97.56** | **98.33** | **98.04** | **98.14** | **98.02** |

Table 24: In- and cross-domain settings of classification **based on the unified 1-D ResNet encoder**. For in-domain setting, we pre-train and fine-tune on the same dataset: Epilepsy. For cross-domain setting, we pre-train on SleepEEG and fine-tune on different domain datasets: Epilepsy, FD-B, Gesture, and EMG.

| | Scenarios | Models | Accuracy (%) | Precision (%) | Recall (%) | F1 (%) | Avg (%) |
|---|---|---|---|---|---|---|---|
| In-Domain | | Random init. | 89.83 | 92.13 | 74.47 | 79.59 | 84.00 |
| | Epilepsy ↓ Epilepsy | TS2vec [55] | 92.33 | 94.53 | 81.11 | 86.33 | 88.58 |
| | | CoST[46] | 92.35 | 94.73 | 81.16 | 85.92 | 88.54 |
| | | LaST [42] | - | - | - | - | - |
| | | TST [56] | 80.89 | 90.38 | 51.73 | 48.01 | 67.75 |
| | | Ti-MAE [21] | 80.34 | 90.16 | 50.33 | 45.20 | 66.51 |
| | | TF-C [57] | 93.96 | 94.87 | 85.82 | 89.46 | 91.03 |
| | | **SimMTM** | **94.75** | **95.60** | **89.93** | **91.41** | **92.92** |
| Cross-Domain | | Random init. | 89.83 | 92.13 | 74.47 | 79.59 | 84.00 |
| | SleepEEG ↓ Epilepsy | TS2vec [55] | 94.46 | 91.99 | 90.28 | 91.10 | 91.95 |
| | | CoST[46] | 93.66 | 91.39 | 88.08 | 89.60 | 90.68 |
| | | LaST [42] | - | - | - | - | - |
| | | TST [56] | 82.89 | 86.15 | 79.02 | 80.44 | 82.13 |
| | | Ti-MAE [21] | 73.45 | 72.56 | 65.34 | 77.20 | 72.14 |
| | | TF-C [57] | 94.95 | **94.56** | 89.08 | 91.49 | 92.52 |
| | | **SimMTM** | **95.49** | 93.36 | **92.28** | **92.81** | **93.49** |
| | | Random init. | 47.36 | 48.29 | 52.35 | 49.11 | 49.28 |
| | SleepEEG ↓ FD-B | TS2vec [55] | 60.74 | 59.60 | 64.27 | 61.07 | 61.42 |
| | | CoST[46] | 54.82 | 51.92 | 63.30 | 54.34 | 56.09 |
| | | LaST [42] | - | - | - | - | - |
| | | TST [56] | 65.57 | 70.05 | 67.57 | 64.41 | 66.90 |
| | | Ti-MAE [21] | 67.98 | 62.83 | 64.45 | 63.36 | 64.66 |
| | | TF-C [57] | 69.38 | **75.59** | 72.02 | 74.87 | 72.97 |
| | | **SimMTM** | **69.40** | 74.18 | **76.41** | **75.11** | **73.78** |
| | | Random Init. | 42.19 | 47.51 | 49.63 | 48.86 | 47.05 |
| | SleepEEG ↓ Gesture | TS2vec [55] | 73.33 | 70.88 | 73.33 | 71.56 | 72.27 |
| | | CoST[46] | 73.33 | 74.37 | 73.33 | 71.16 | 73.04 |
| | | LaST [42] | - | - | - | - | - |
| | | TST [56] | 75.12 | 76.05 | 67.74 | 73.24 | 73.04 |
| | | Ti-MAE [21] | 75.54 | 69.32 | 72.42 | 69.32 | 71.65 |
| | | TF-C [57] | 76.42 | 77.31 | 74.29 | 75.72 | 75.94 |
| | | **SimMTM** | **80.00** | **79.03** | **80.00** | **78.67** | **79.43** |
| | | Random init. | 77.80 | 59.09 | 66.67 | 62.38 | 66.49 |
| | SleepEEG ↓ EMG | TS2vec [55] | 80.92 | 69.63 | 67.65 | 67.90 | 71.52 |
| | | CoST[46] | 73.17 | 70.47 | 69.84 | 70.00 | 70.87 |
| | | LaST [42] | - | - | - | - | - |
| | | TST [56] | 75.89 | 74.67 | 80.66 | 78.48 | 77.43 |
| | | Ti-MAE [21] | 63.52 | 67.77 | 70.55 | 58.32 | 65.04 |
| | | TF-C [57] | 81.71 | 72.65 | 81.59 | 76.83 | 78.20 |
| | | **SimMTM** | **97.56** | **98.33** | **98.04** | **98.14** | **98.02** |

Table 25: Full ablation studies for in-domain and cross-domain settings of classification. Under the *Avg* metric, the standard deviations of SimMTM are within 0.2% for Epilepsy, within 0.5% for FD-B, within 0.6% for Gesture, and within 0.1% for EMG.

| Scenarios | | Accuracy (%) | Precision (%) | Recall (%) | F1 (%) | Avg (%) |
|---|---|---|---|---|---|---|
| Epilepsy ↓ Epilepsy | Random init. | 89.83 | 92.13 | 74.47 | 79.59 | 84.00 |
| | W/o $\mathcal{L}_{\text{reconstruction}}$ | 93.80 | **96.11** | 86.11 | 89.45 | 91.37 |
| | W/o $\mathcal{L}_{\text{constraint}}$ | 90.99 | 92.81 | 79.86 | 84.13 | 86.95 |
| | **SimMTM** | **94.75** | 95.60 | **89.93** | **91.41** | **92.92** |
| SleepEEG ↓ Epilepsy | Random init. | 89.83 | 92.13 | 74.47 | 79.59 | 84.00 |
| | W/o $\mathcal{L}_{\text{reconstruction}}$ | 94.54 | **93.87** | 88.46 | 90.84 | 91.93 |
| | W/o $\mathcal{L}_{\text{constraint}}$ | 91.73 | 90.57 | 82.21 | 85.53 | 87.51 |
| | **SimMTM** | **95.49** | 93.36 | **92.28** | **92.81** | **93.49** |
| SleepEEG ↓ FD-B | Random init. | 47.36 | 48.29 | 52.35 | 49.11 | 49.28 |
| | W/o $\mathcal{L}_{\text{reconstruction}}$ | 66.11 | 67.97 | 74.70 | 70.01 | 69.70 |
| | W/o $\mathcal{L}_{\text{constraint}}$ | 53.71 | 69.48 | 62.67 | 50.86 | 59.18 |
| | **SimMTM** | **69.40** | **74.18** | **76.41** | **75.11** | **73.78** |
| SleepEEG ↓ Gesture | Random init. | 42.19 | 47.51 | 49.63 | 48.86 | 47.05 |
| | W/o $\mathcal{L}_{\text{reconstruction}}$ | 78.50 | 79.01 | 78.50 | 77.17 | 78.30 |
| | W/o $\mathcal{L}_{\text{constraint}}$ | 76.67 | 74.91 | 76.67 | 74.80 | 75.76 |
| | **SimMTM** | **80.00** | **79.03** | **80.00** | **78.67** | **79.43** |
| SleepEEG ↓ EMG | Random init. | 77.80 | 59.09 | 66.67 | 62.38 | 66.49 |
| | W/o $\mathcal{L}_{\text{reconstruction}}$ | 90.24 | 94.20 | 78.04 | 81.53 | 86.00 |
| | W/o $\mathcal{L}_{\text{constraint}}$ | 85.37 | 89.97 | 69.62 | 70.74 | 78.93 |
| | **SimMTM** | **97.56** | **98.33** | **98.04** | **98.14** | **98.02** |

