# OpenReview forum: "SimMTM: A Simple Pre-Training Framework for Masked Time-Series Modeling"
_NeurIPS.cc/2023/Conference — NeurIPS 2023 spotlight_

### Official Review · Reviewer_5Qee · 2023-07-04

**Soundness:** 3 good
**Presentation:** 3 good
**Contribution:** 3 good
**Rating:** 7
**Confidence:** 3

**Summary:**

This paper presents SimMTM, a simple pre-training framework for masked time-series modeling. The core idea is to train the model to reconstruct the time series by aggregating information from multiple masked series rather than one single masked series. While the temporal pattern is destroyed in a single masked series, multiple masked series contain complementary information, making the reconstruction process more accessible. The aggregation of point-wise representations is weighted by their series-wise similarities. The proposed loss adds the reconstruction loss with a contrastive loss on the series representations. The author provides extensive empirical studies on both time series classification and time series forecasting tasks to evaluate the learned representations.

**Strengths:**

The proposed method is sound and well-motivated. Reconstructing the original time series from multiple randomly masked series is novel, in my opinion. The write-up is easy to follow. The overview figure is also illustrative and helps the understanding. The empirical evaluation is comprehensive and convincing.

**Weaknesses:**

1. In Eq 4, the sum over the variable $s'$ is confusing without carefully reading the text below it. In my opinion, it would be more clear to have the sum over $z'$ and have $s' = Projector(z')$ on the right.

2. Eq 7 is not precise. I understand that the authors mean the intra-similarity of samples should be maximized. However,  the sets before and after $~$ are the same, and therefore they are already "close". Please considering revise eq 7.

**Questions:**

The paper says, "For each time series, the reconstruction is not only based on its own masked series." I wonder how much does the method benefit from aggregating representations from other time series?



**Limitations:**

Yes

---

> ### Author Rebuttal · Authors · 2023-08-08
>
> We sincerely thank Reviewer 5Qee for providing a meaningful review and insightful suggestions.
>
> #### **About the Weaknesses**
>
> **Q1**: The confusing of $Eq. (4)$.
>
> Thank you for your suggestion. Your understanding is correct. We will provide a clearer description and modify $Eq. (4)$ as follows:
>
> - Rephrase $\underline{\text{lines 153-154 in the main text}}$ as:
>
>      where $\mathbf{s}^{\prime}$ represents series-wise representation, while $\mathbf{z}^\prime$ represents the corresponding point-wise representation, where $\mathbf{s}^{\prime} = \operatorname{Projector}(\mathbf{z}^{\prime})$ and $\widehat{\mathbf{z}}\_{i}\in\mathbb{R}^{L\times d_{\text{model}}}$ is the extracted point-wise representation.
>
> - Modify $\underline{\text{Eq. (4) in the main text}}$ to:
> $$
> \begin{equation}
> \begin{split}
>     & \mathbf{s}^{\prime} = \text{Projector}(\mathbf{z}^{\prime}), \newline
>     & \widehat{\mathbf{z}}\_{i} = \sum\_{\mathbf{s}^{\prime} \in \mathcal{S}\backslash\\{\mathbf{s}\_{i}\\}}\frac{\text{exp}({\mathbf{R}\_{\mathbf{s}\_{i},{\mathbf{s}^\prime}}/\tau})}{\sum\_{{\mathbf{s}^\prime}^{\prime} \in \mathcal{S}\backslash\\{\mathbf{s}\_{i}\\}} \text{exp}({\mathbf{R}\_{\mathbf{s}\_{i},{\mathbf{s}^\prime}^\prime}/\tau})}\mathbf{z}^\prime
> \end{split}
> \end{equation}
> $$
>
> **Q2**: $Eq. (7)$ is not precise.
>
> Thanks for your suggestion, and we will rewrite $Eq. (7)$ to better represent positive pairs and negative pairs as follows:
>
> $$
> \begin{equation}
> \begin{split}
>     & \text{Positive pairs:}~~\left(\mathbf{s}\_{i}, \mathbf{s}\_{i}^{+} \right), \mathbf{s}\_{i}^{+} \in \{\overline{\mathbf{s}}\_{i}^{j}\}\_{j=1}^{M}, \newline
>     & \text{Negative pairs:}~\left(\mathbf{s}\_{i}, \mathbf{s}\_{i}^{-} \right),
>     \mathbf{s}\_{i}^{-} \in \{\mathbf{s}\_{k}\} \cup \{\overline{\mathbf{s}}\_{k}^{j}\}\_{j=1}^{M}, i\neq k
> \end{split}
> \end{equation}
> $$
>
> #### **About the Questions**
>
> **Q1**: What is the benefit of aggregating representations from other time series?
>
> As we stated in $\underline{\text{lines 155-159 in the main text}}$, the aggregation of representations from other time series requires the model to suppress the interference of less-related noise series and precisely learn similar representations for both the masked and the original series, namely guiding the model to learn the manifold structure better.
>
> To further verify the benefit, we also conducted a comparison experiment as follows, where "Own Masked Series" represents aggregating representations from their own masked series, and "All Masked Series" means aggregating representations from their own and other masked series. We present the averaged MSE/MAE from 4 different forecasting horizons {96,192,336,720\} based on the past 336 time points for the in-domain forecasting tasks. We also record the Accuracy (%) score for classification tasks as follows:
>
> - Forecasting Tasks
>     | Average MSE/MAE|   Own Masked Series | All Masked Series (Ours) |
>     | :-------------:| :------------: | :------------: |
>     | ETTh1          |   0.407/0.430  |   0.404/0.428  |
>     | ETTh2          |   0.348/0.393  |   0.348/0.391  |
>     | ETTm1          |   0.348/0.380  |   0.340/0.379  |
>     | ETTm2          |   0.263/0.318  |   0.260/0.318  |
>     | Avg            |   0.342/0.380  |   **0.338**/**0.379**  |
>
> - Classification Tasks
>     |    Accuracy (%)   |   Own Masked Series | All Masked Series (Ours) |
>     | :------------------:| :-----------------: | :----------------------: |
>     | Epilepsy → Epilepsy |   94.18             |             94.75        |
>     | SleepEEG → Epilepsy |   95.49             |             95.49        |
>     | SleepEEG → FD-B     |   67.01             |             69.40        |
>     | SleepEEG → Gesture  |   77.08             |             80.00        |
>     | SleepEEG → EMG      |   92.24             |             97.56        |
>     | Avg                 |   85.20             |           **87.44**      |
>
> The experimental results show that "All Masked Series" performs consistently better than "Own Masked Series" in forecasting and classification tasks, which proves that using both own and other less-related noise series for reconstruction is meaningful.

---

### Official Review · Reviewer_L1Yx · 2023-07-05

**Soundness:** 3 good
**Presentation:** 3 good
**Contribution:** 3 good
**Rating:** 8
**Confidence:** 3

**Summary:**

Pre-training models have been using self-supervised learning in various fields (NLP, Vision). However, masked modeling, a representative method of self-supervised learning, was difficult to apply to time-series tasks. Randomly masked data is difficult to show good performance because semantic information of time series including temporal changes is broken. Thus, SimMTM recovers the masked point in time through weighted aggregation of multiple neighbors outside the manifold. This not only restores simple, but also reconstructs complementarily, leading to good performance. SimMTM performs forecasting and classification (in-domain, cross-domain) on various datasets and shows good performance.

**Strengths:**

1. They explained the problem of masked modeling in time-series data well from the time-series point of view.
2. Also, to apply masked modeling to time-series data, the characteristics of time-series were well utilized. In other words, masked modeling was not simply applied to time-series, but the characteristics of time-series data were well understood and applied accordingly.
3. Experiments were conducted on various tasks of the pre-training model, and remarkable performance improvements were achieved.

**Weaknesses:**

It seems that there is a part that does not match the notation in the formula and Figure 2.

**Questions:**

Regarding weakness, I don't know what $z'$ means in Eq.4 and line 154. No explanation of $s'$ in "$z'$ represents the corresponding point-wise representation of $s'$". Does $s'$ mean series-wise representations $S$?

**Limitations:**

In fact, there is no limit to this work

---

> ### Author Rebuttal · Authors · 2023-08-08
>
> We would like to sincerely thank Reviewer L1Yx for providing an insightful review.
>
> #### **About the Weaknesses**
>
> **Q1**: A part that does not match the notation in the formula and Fig. 2.
>
> Thank you for your careful reading. To facilitate summation operations and distinguish different sample representations, we have introduced new symbols, $\mathbf{z}^{\prime}$ and $\mathbf{s}^{\prime}$.
>
> - $\mathbf{z}^{\prime}$ represents the point-wise representation of time series.
> - $\mathbf{s}^{\prime}$ represents the series-wise representation of time series.
>
> #### **About the Questions**
>
> **Q1**: The explanation of $\mathbf{z}^{\prime}$ and $\mathbf{s}^{\prime}$.
>
> Thank you for the constructive reviews, your understanding is correct. We will modify the description and $Eq. (4)$ in the original text as follows:
>
> - Rephrase $\underline{\text{lines 153-154 in the main text}}$ as:
>
>      where $\mathbf{s}^{\prime}$ represents series-wise representation, while $\mathbf{z}^\prime$ represents the corresponding point-wise representation, where $\mathbf{s}^{\prime} = \operatorname{Projector}(\mathbf{z}^{\prime})$ and $\widehat{\mathbf{z}}\_{i}\in\mathbb{R}^{L\times d_{\text{model}}}$ is the extracted point-wise representation.
>
> - Modify $\underline{\text{Eq. (4) in the main text}}$ to:
> $$
> \begin{equation}
> \begin{split}
>     & \mathbf{s}^{\prime} = \text{Projector}(\mathbf{z}^{\prime}), \newline
>     & \widehat{\mathbf{z}}\_{i} = \sum\_{\mathbf{s}^{\prime} \in \mathcal{S}\backslash\\{\mathbf{s}\_{i}\\}}\frac{\text{exp}({\mathbf{R}\_{\mathbf{s}\_{i},{\mathbf{s}^\prime}}/\tau})}{\sum\_{{\mathbf{s}^\prime}^{\prime} \in \mathcal{S}\backslash\\{\mathbf{s}\_{i}\\}} \text{exp}({\mathbf{R}\_{\mathbf{s}\_{i},{\mathbf{s}^\prime}^\prime}/\tau})}\mathbf{z}^\prime
> \end{split}
> \end{equation}
> $$

---

> > ### Comment · Reviewer_L1Yx · 2023-08-18
> >
> > Thank you for solving my question.

---

### Official Review · Reviewer_ZNHw · 2023-07-06

**Soundness:** 3 good
**Presentation:** 2 fair
**Contribution:** 3 good
**Rating:** 5
**Confidence:** 3

**Summary:**

The authors propose SimMTM, a novel representation learning method for time-series data based on random masking. They experimentally verify the effectiveness of the proposed method from both in-domain and cross-domain perspectives. Comprehensive comparative experiments with existing methods are conducted, demonstrating the effectiveness of the proposed approach.

**Strengths:**

The authors have conducted extensive comparative experiments with existing methods and quantitatively demonstrated the effectiveness of the proposed method. In addition to in-domain generalization, they also focus on cross-domain generalization. This verification is indispensable for validating the effectiveness of deep representation learning assuming large-scale open data.

**Weaknesses:**

**Presentation quality**

The current presentation style of this paper is confusing. Particularly, there is a lack of consistency in the representation of formulas and symbols, making it very difficult to grasp the overall logic of the paper. Some specific examples include:

1. In Equation 3, the symbol $\times$ is used to denote both the Cartesian product and scalar multiplication when defining the domain of $R$.
2. The transposition notation in Equation 3 is counterintuitive as $\mathbf{s}$ is defined as a row vector.
3. Equation 7 is overly verbose and exacerbates the difficulty in reading, as it does not contribute to the subsequent. It would be sufficient to simply declare that only the situation where $i = k$ is treated as positive pairs.
4. While $\mathbf{s}$ is defined as a single vector, $\mathbf{s}^+$ is defined as a set of vectors.

Some other points that I could not make from the paper are discussed below in the Questions section.

**Comparison with prior literature**

The comparison with prior literature is insufficient. For example, references [1, 2] can be cited as representative examples of representation learning methods for deep neural networks under time-series data. In particular, a model that generalized the method in [2] as a generalized contrastive learning method not limited to the time-series domain was proposed later. Both [2] and [3] have superior theoretical guarantees for identifiability from the perspective of nonlinear ICA. The authors should clearly indicate the proposed model's advantages over these approaches.

**References**

1. Hyvarinen, A., & Morioka, H. (2016). Unsupervised feature extraction by time-contrastive learning and nonlinear ICA. *Advances in neural information processing systems*, *29*.
2. Oord, A. V. D., Li, Y., & Vinyals, O. (2018). Representation learning with contrastive predictive coding. *arXiv preprint arXiv:1807.03748*.
3. Hyvarinen, A., Sasaki, H., & Turner, R. (2019, April). Nonlinear ICA using auxiliary variables and generalized contrastive learning. In *The 22nd International Conference on Artificial Intelligence and Statistics* (pp. 859-868). PMLR.

**Questions:**

- What is the definition of $\operatorname{Encoder}(\mathcal{X})$? Typically, a function $f \colon X \rightarrow Y$ can use all the information in $X$ to calculate $Y$, but it seems that the Encoder function defined here applies the same function in parallel to some dimension. However, it is not clear from the text which dimension this parallel processing applies to.
For example, there would be at least two possibilities: $\operatorname{Encoder}(\mathcal{X})$ is kind of a “syntax suger” for either $\bigcup\_i \operatorname{Encoder}( \\{x\_i\\} \cup \\{\bar{x}\_i^j \\}\_{j} )$ or $\bigcup\_i ( \\{ \operatorname{Encoder}(x\_i) \\} \cup \\{ \operatorname{Encoder}(\bar{x}\_i^j) \\}\_{j} )$. The same can be said about the definitions of the Projector and Decoder.
- Although I understood the training pipeline of the proposed method, I had questions regarding its behavior at inference time. Does the proposed method compute internal representations from the true sequence and $M$ randomly masked sequences at inference time, similar to training time? Or does it only use the true sequence without random masking at inference time? Additionally, how is training conducted during the fine-tuning stage in the pipeline?
- The authors mention the analysis of a quantity called the CKA value in line 262 of Section 4.4, but it is unclear what this quantity actually is. While there is an explanation of Pre-training/Fine-tuning CKA in the caption of Table 5, if the CKA value mentioned in the main text refers to these, its meaning should be clearly explained in the text as well. Moreover, I did not understand why a small $| \Delta_{\mathrm{CKA}} |$ leads to “acquiring adaptive representations for different tasks” (l.264). In principle, it should be possible to drastically change the internal representation by fine-tuning while maintaining the degree of change in the internal representation along the evolution of the layers. Suppose $x_\mathrm{first}$ and $x_\mathrm{last}$ be the representation in the first and the last layer at the pre-training phase, respectively. We can consider one idealistic situation that the representations after fine-tuning are given as $y_{\mathrm{first}} = x_{\mathrm{first}} + \delta_{\mathrm{task}}$ and $y_{\mathrm{last}} = x_{\mathrm{last}} + \delta_{\mathrm{task}}$ with a task-dependent fixed vector $\delta$. In such a situation, the values of Pre-training/Fine-tuning CKA would be nearly the same, and the difference in CKA values would go to zero by construction. However, the internal representations drastically change depending on the task choice. In this context, I believe there is a logical gap in the authors' argument.
- In masked modeling-based representation learning, the proposed method uses the true sequence without masking, along with the input. This setup is generally considered to make it more challenging to avoid trivial local solutions that merely output the input as is. Can it be said that such a phenomenon does not occur? Moreover, if a trivial local solution can be avoided, what factors of the proposed method enable this?
- As a naive control condition for the idea of using multiple different random masks, it is possible to consider the variant of using only one random mask that reduces the masked portion $r$, or takes the position of the mask to have a specific rules that is not completely iid. Can the proposed method be claimed to be effective against such variants? In particular, the model structure of using multiple random masks is expected to have at least some overhead in terms of training/inference computation time compared to such simple control conditions.

**Limitations:**

The limitations of this paper are adequately discussed in the appendix. Furthermore, the effectiveness of the proposed method can be confirmed from various perspectives through exhaustive comparative experiments. However, as mentioned earlier, there are major concerns primarily about the presentation of the paper. These are points that affect the reproducibility of the results and the authors' claims, so it would be necessary to resolve all problems of unclear logical progression for this paper to be accepted.

---

> ### Author Rebuttal · Authors · 2023-08-09
>
> We sincerely thank Reviewer ZNHw for the detailed and insightful suggestions.
>
> #### **About Presentation Quality**
>
> (1) We will rephrase $Eq. (3)$ as $\mathbf{R}=\text{Sim}(\mathcal{S})\in\mathbb{R}^{D\times D},D=N(M+1),$
>
> (2) $\mathbf{R}_{\text{u},\text{v}}$ compute the cosine distance via transposition in $Eq. (3)$, resulting in a value that represents the similarity between vectors $\text{u}$ and $\text{v}$.
>
> (3) We will remove $Eq. (7)$ in the main text.
>
> (4) We will modify $\text{s}^+$ to $\mathcal{S}^+$ as a positive series set.
>
> #### **About Comparison with Prior Literature**
>
> **Q1**: The comparison with prior literature is insufficient.
>
> We have compared SimMTM with six competitive state-of-the-art baselines in the main text, which are representative and recently published in the time series domain, including Ti-MAE(2023), TST(2021), LaST(2022), TF-C(2022), etc.
>
> As per reviewer's request, we have reimplemented and compared the generalized methods you suggested in $\underline{\text{Table~1, 2 in the global response PDF}}$. SimMTM performs best and achieves 15.5%, 11.5%, and 10.6% average MSE reduction compared to TCL(2016), CPC(2018), and PCL(2019) in forecasting tasks, improves 3.33% average accuracy compared to PCL(2019) in classification tasks.
>
> #### **About the Questions**
>
> **Q1**: What is the definition of $\text{Encoder}(\mathcal{X})$ and how to apply it?
>
> In the previous formulation, we organized time series and its masked series along with batch dimension, which is a conventional usage in pre-training. Thus, as you pointed out, $\text{Encoder}(\mathcal{X})=\bigcup_i(\{\text{Encoder}(\text{x}\_i)\}\cup\{\text{Encoder}(\overline{\text{x}}\_i^j)\_j\})$, which means $\text{Encoder}$ will process the input series separately.
>
> **Q2**: Understanding the training pipeline of SimMTM.
>
> SimMTM follows the standard pre-training and fine-tuning paradigm, including three stages: pre-training, fine-tuning, and inference. SimMTM mainly focuses on modeling time series representations in pre-training. Notably, masking is only applied in pre-training, not fine-tuning and inference.
> - In fine-tuning, the pre-trained Encoder is retained while Projector and Decoder are removed. Different task heads are appended to the pre-trained Encoder and fine-tune all parameters for ten epochs to validate the performance. Note that we organize data into batches for fine-tuning without masking.
> - In inference, the model's parameters remain unchanged, and the fine-tuned model is directly used for inference. A single series without masking will be input to Encoder with the specific head.
>
> **Q3**: What is $\text{CKA}$ and why a small pre-training/fine-tuning $|\Delta_\text{CKA}|$ acquire adaptive representations for different tasks?
>
> (1) $\text{CKA}$ can be used to measure the representation-learning property of deep models.
>
> Centered Kernel Alignment ($\text{CKA}$) is a statistical measure between representations that identify correspondences in models with different initializations. We can use it to measure the representation-learning property of deep models by calculating $\text{CKA}$ between a model's first and last layer representations. A larger $\text{CKA}$ indicates the model tends to learn high-level representations, which is widely used in previous works [1,2,3].
>
> (2) $|\Delta_\text{CKA}|$ can measure the difference in representation-learning property between pre-training and fine-tuning models.
>
> Suppose both pre-training and fine-tuning models hold the same high-level or low-level representation-learning property, these two models will present the close $\text{CKA}$ values, corresponding to smaller $|\Delta_\text{CKA}|$. Thus, we use the $|\Delta_\text{CKA}|$ to measure the gap between pre-training and fine-tuning models. In other words, **we do not attempt to use $|\Delta_\text{CKA}|$ to measure the change of representations but use it to quantify the change of model representation-learning property**.
>
> [1] Xie,et al, Revealing the Dark Secrets of Masked Image Modeling. CVPR, 2022.
> [2] Kornblith,et al, Similarity of Neural Network Representations Revisited. ICML, 2019.
> [3] Wu,et al, TimesNet: Temporal 2D-Variation Modeling for General Time Series Analysis. ICLR, 2023.
>
> **Q4**: Why the usage of true series in SimMTM will not lead to trivial local solutions?
>
> Note that SimMTM reconstructed point-wise series representations based on series-wise representation similarity. Although the true series-wise representation $\text{s}\_i$ undergo similarity calculation in $Eq. (2)$, we only use the point-wise representations of $\text{z}^\prime$ for aggregation (corresponding to $\text{s}^\prime \in \mathcal{S}\backslash\{\text{s}\_i\}$). It implies that the reconstruction of SimMTM solely relies on the series-wise similarity values among the true and other series, disregarding the specific point-wise values of the true series. It is not simply reconstructing the original series with the true series, so trivial local solutions will not occur.
>
> **Q5**: Try variants of masking rules.
>
> The variant you mentioned, only using one masked series with a lower masked ratio can perform well but cannot beat multiple masked series settings. (As shown in $\underline{\text{Fig. 5 in the main text}}$). Further, we have analyzed the relationship between masked ratio and masked numbers in Fig. 5, where a reasonable balance between masked ratio and masked numbers is critical.
>
> As you requested, we also compared different masked rules for ETTh1, including:
> - Fixed position: Masking tail or head
> - Random position: Masking large, small segment or points
>
> The experiments are in $\underline{\text{Table~3 in the global response PDF}}$. Results show random masking is superior to fixed masking (Small Segment > MaskedTail > Masked Head). The difference is insignificant among random masking.
>
> As mentioned in **Q2**, masked modeling involves only pre-training, so there is no computational overhead problem in fine-tuning and inference.

---

> > ### Comment · Reviewer_ZNHw · 2023-08-22
> > **Thank you for your response**
> >
> > Thank you for responding to my questions and concerns.
> >
> > Regarding presentation quality, I think the paper will be easier to read if revisions are made based on the comments. Thanks for adding the experimental comparison to the previous studies I have pointed out. After confirming the experimental results, I acknowledge the experimental advantage of this paper over these papers.
> >
> > About questions
> >
> > **Q1&Q2** I understood the procedure of this work. Thank you for your response.
> >
> > **Q3** I am still not fully convinced. Regarding CKA, the authors stated as
> >
> > > Centered Kernel Alignment (CKA) is a statistical measure between representations that identify correspondences in models with different initializations. We can use it to measure the representation-learning property of deep models by calculating between a model's first and last layer representations. A larger indicates the model tends to learn high-level representations, which is widely used in previous works [1,2,3].
> >
> > First, CKA itself is a general metric that evaluates the similarity between two representations, not limited to the models with different initialization or layers. I acknowledge that CKA itself is widely used to evaluate the similarity of model representations. However, I was still not convinced by the authors' statement that evaluating the CKA of the first and last layers of a model can tell whether the model has acquired "high-level representations." It seems to me that high CKA simply means high representaion similarity between layers, but how could this lead to the acquisition of "high-level representaion” as an entire model?
> >
> > **Q4** I understood the authors' point.
> >
> > **Q5**
> > > As mentioned in Q2, masked modeling involves only pre-training, so there is no computational overhead problem in fine-tuning and inference.
> >
> > Thank you for making clear this point. However, I asked about *some overhead in terms of both training/inference computation time.* The computational overhead of training time because of using multiple masking should be clearly documented in the manuscript.
> >
> > Overall, the additional experiments and notational revisions made the paper's contribution clear. But, I still have a bit concern on the interpretation of CKA studies. Therefore, let me raise my score to 5.

---

> > > ### Author Response · Authors · 2023-08-22
> > > **Thanks for Your Response and Raising the Score**
> > >
> > > We thank Reviewer ZNHw for providing a detailed valuable rebuttal review and feedback, which enables us to understand your concerns clearly.
> > >
> > > We are delighted to further clarify the remaining two questions:
> > >
> > > **(1)** About the CKA: we adopt this concept following the usage of the previous works [1, 2]:
> > >
> > > - Xie et al. [1] use the CKA to measure the attention similarity among different layers in vision Transformer, specially "measure the representation diversity in the model."
> > >
> > > - TimesNet [2], which also focuses on time series (**closer to our paper**), adopts the CKA similarity between bottom and top layers to define "high-level" and "low-level" representations (or tasks) quantitively ($\underline{\text{Section 4.6 of their official paper}}$).
> > >
> > > Intuitively, since the bottom layer representations are usually be viewed as containing the "low-level" or detailed information, a smaller CKA similarity means the top layer contains the different information from the bottom layer, so-called "high-level" or abstract information.
> > >
> > > Many thanks for your valuable question. We will rephrase this part in every detail by adding new explanations and proper citations to make the concept more clear.
> > >
> > > **(2)** About the training overhead: Thanks for pointing out this item. We will discuss the training overhead in the $\underline{\text{Conclusion}}$ section as a limitation and place it in the future work.
> > >
> > > Many thanks for your dedication! We guarantee to resolve all the writing issues and include all the mentioned updates in the final version.
> > >
> > > =====
> > >
> > > [1] Xie,et al, Revealing the Dark Secrets of Masked Image Modeling. CVPR, 2022.
> > >
> > > [2] Wu,et al, TimesNet: Temporal 2D-Variation Modeling for General Time Series Analysis. ICLR, 2023.

---

> ### Author Response · Authors · 2023-08-19
> **Request of Reviewer's attention and feedback**
>
> Dear Reviewer,
>
> We kindly remind you that it has only 2 days until the Reviewer-author discussion ends. Please let us know if our response has addressed your concerns. Due to the word limit of Rebuttal, we will be happy to answer your concerns further or deal with any additional issues/questions in the discussion period.
>
> Following your suggestion, we have answered your concerns and improved the paper in the following aspects:
>
> - We **have rewritten or modified some formulas and corrected some writing errors** to improve presentation quality.
> - We **have compared SimMTM and three generalized representation learning methods you mentioned TCL(2016), CPC(2018), and PCL(2019)**. We further demonstrate the state-of-the-art performance of SimMTM.
> - We **explained the training pip-line of pre-training and fine-tuning**, **the definition of Encoder**, **What is $\text{CKA}$, and why we use $|\Delta_{\text{CKA}}|$ to measure the difference in representation-learning property between pre-training and fine-tuning models**.
> - We further **analyzed the relationship between masked ratios and numbers** and **compared the effect of different masked rules**.
>
> Since we cannot submit the revised version during the reviewing phase, we guarantee to resolve all the
> writing issues and include all the above updates in the revised paper.
>
> Thanks again for your valuable review. We are looking forward to your reply.

---

> ### Author Response · Authors · 2023-08-21
> **We are anticipating your feedback.**
>
> Dear Reviewer ZNHw,
>
> Thanks again for your valuable and constructive review, which has inspired us to improve our paper further substantially.
>
> Following your suggestions, we have modified some formulas and writing errors to improve presentation quality, compared the three generalized representation learning methods you mentioned, analyzed the relationship between masked ratios and numbers, compared the effect of different masked rules, and discussed all your mentioned weaknesses in every detail. We do our best to solve your concerns in the limited time and characters.
>
> We hope that this new version has addressed your concerns to your satisfaction. We eagerly await your reply and are happy to answer any further questions. **We kindly remind you that the reviewer-author discussion phase will end by Aug 21st at 1 pm EDT, just 4 hours left. After that, we may not have a chance to respond to your comments**.
>
> Sincere thanks for your dedication!
>
> Authors

---

### Official Review · Reviewer_xbHA · 2023-07-07

**Soundness:** 2 fair
**Presentation:** 4 excellent
**Contribution:** 3 good
**Rating:** 5
**Confidence:** 4

**Summary:**

This paper proposes a simple pre-training framework for masked time-series modeling. Instead of reconstructing origin data directly, which is unsuitable for time series, this paper recovers masked time points by the weighted aggregation of multiple neighbors outside the manifold. The reviewer appreciates the novelty method proposed, but is still concerned about some results.

**Strengths:**

1) The method is of novelty. The multi-level aggregation and representation method is appealing.
2) Experimentally, both high-/low-level tasks benefit from this method.

**Weaknesses:**

1. The experimental results need more justification (see Questions).
2. The relationship with manifold learning needs more description. What is the progress in this aspect?
3. The masked parts in Figure 1 and Figure 1 in Supplement material should be annotated and more results should be listed. A well-trained network usually predicts low-frequency results. Why the result of TST in the right-upper part in Figure 1 in Supplement material contains too many high-frequency parts?


**Questions:**

In terms of experimental results, why Random init. outperforms TF-C in Table 2? Does SimMTM use a stronger baseline? Is it an unfair comparison? Maybe TF-C or other methods will outperform SimMTM with the same baseline. It seems that the model learns directly aggregate information from different masked series, which learns more temporal information than other semantic information.


**Limitations:**

The authors should discuss more implications about using large-scale pre-training in forecasting in terms of fairness.

---

> ### Author Rebuttal · Authors · 2023-08-08
>
> We would like to sincerely thank Reviewer xbHA for providing an insightful review.
>
> #### **About the Weaknesses**
>
> **Q1**: The relationship with manifold learning needs more description. What is the progress in this aspect?
>
> Masked modeling is a mainstream paradigm in self-supervised pre-training. However, the temporal information of the time series is significantly distorted as the masked ratio increases, making the reconstruction task too difficult to guide representation learning. SimMTM firstly introduces the core idea of neighborhood aggregation from manifold learning into self-supervised time series representation learning by reconstructing the original time series using multiple neighborhood masked series.
>
> Numerous experimental results further demonstrate the importance of introducing the idea of neighborhood aggregation from manifold learning into self-supervised representation learning of time series.
> - SimMTM achieves 8.9% average MSE reduction (0.335→0.305) and 4.6% average MAE reduction (0.445→0.329) compared to the advanced masked modeling baseline Ti-MAE (2023) in in-domain forecasting benchmarks ($\underline{\text{Table 3 in the main text}}$), which reconstructs the original time series based on standard masked modeling.
> - The averaged accuracy of SimMTM remarkably surpasses previous state-of-the-art TF-C (2022) (87.44% vs. 83.29%), as shown in $\underline{\text{Table 4 in the main text}}$.
>
> **Q2**: Why the results of TST contains too many high-frequency parts?
>
> Masking will make the input series temporally irregular, bringing noise to the reconstruction, thereby leading to high-frequency results. Following your suggestion, we have included more showcases in $\underline{\text{Figure 1 in the global response PDF}}$.
>
> SimMTM reconstructs the original time series by aggregating multiple neighboring masked series. The information from multiple neighboring masked time series is aggregated reasonably and complements each other. Multiple neighboring aggregating brings a more stable reconstruction effect, which is more beneficial for the representation learning of time series.
>
> #### **About the Questions**
>
> **Q1**: Why Random init. outperforms TF-C?
>
> We have mentioned the particularity of TF-C and LaST in $\underline{\text{lines 207-209 in the main text}}$. TF-C is closely related to its model structure. Thus we do not unify the Encoder of TF-C in the original submission.
>
> In addition, TF-C primarily focuses on time series classification tasks, not forecasting tasks. Specifically, TF-C, which is based on time and frequency series-wise representation contrastive learning, is more suitable for series-wise classification tasks than point-wise forecasting tasks.
>
> To further validate, we attempted to replace the dual-tower TF-C (CNN) with TF-C (Transformer). We show the averaged MSE/MAE from 4 different forecasting lengths {96,192,336,720} based on the past 336 time points. Results showed that TF-C (Transformer) outperformed TF-C (CNN) in forecasting tasks but still significantly lagged behind SimMTM. Note that even if we change each part of it the same as SimMTM (Transformer), its unique dual-tower design cannot be strictly consistent with SimMTM.
>
> - Forecasting tasks in the in-domain setting
>     |Average MSE/MAE|TF-C (CNN)|TF-C (Transformer)|SimMTM
>     |:-:|:-:|:-:|:-:|
>     |ETTh1|0.637/0.638|0.492/0.481|0.404/0.428
>     |ETTh2|0.398/0.398|0.401/0.437|0.348/0.391
>     |ETTm1|0.744/0.652|0.466/0.445|0.340/0.379
>     |ETTm2|1.755/0.947|0.295/0.341|0.260/0.318
>
> **Q2**: Does SimMTM use a stronger baseline? Is it an unfair comparison?
>
> Ensuring complete consistency in the backbone used by all baselines is often unrealizable due to the high dependence between model architecture and the pre-training method in this domain. Existing work also commonly faces this problem [1,2,3].
>
> We have made a great effort to ensure a fair comparison. For all baselines and all tasks, we have conducted both experiments: using a unified backbone and the specific backbone proposed in their original papers. The comprehensive experiments and descriptions have been presented in $\underline{\text{lines 203-211 in the main text}}$, $\underline{\text{Table 7, 8 in Supplementary Materials}}$.
>
> [1] Yue, et al., TS2Vec: towards universal representation of time series. AAAI, 2022.
> [2] Zhang, et al., Self-supervised contrastive pre-training for time series via time-frequency consistency. NeurIPS, 2022.
> [3] Nie, et al., A time series is worth 64 words: long-term forecasting with transformers. ICLR, 2023.
>
> **Q3**: The model learns more temporal information than other semantic information.
>
>  As stated in $\underline{\text{lines 5-6 in abstract}}$, the semantic information of time series is mainly concentrated in temporal variations. Thus, the detailed semantic information can be learned by aggravating multiple time series for reconstruction.
>
>  In addition, the aggregation process also relies on the similarity among series-wise representations from multiple time series, which makes the model also learn the global semantic information.
>
> #### **About the Limitations**
>
> **Q1**: The authors should discuss more implications about using large-scale pre-training in forecasting in terms of fairness.
>
> We pay great attention to the issue of fair comparison.
> - We conduct two types of experiments for all baselines and all tasks: using a unified backbone and the specific backbone proposed in their original papers.
> - We strictly separate training, validation, and testing for all data to prevent data leakage issues.

---

> > ### Comment · Reviewer_xbHA · 2023-08-19
> > **Thanks for the response**
> >
> > Thanks for the response. At this stage, I will keep my score unchanged.

---

> > > ### Comment · Area_Chair_mxUn · 2023-08-21
> > >
> > > Thanks for the updated opinion reviewer xbHA, however, could you give more detailed reasoning as why you have your score unchanged? The authors have made an effort to respond to your comments, so please at least point out which part is not satisfying enough.

---

### Author Rebuttal · Authors · 2023-08-09

## Summary of Revisions and Global Response

We sincerely thank all the reviewers for their insightful reviews and valuable comments, which are instructive for improving our paper further.

Since the standard masked modeling seriously ruins vital temporal variations of time series, this paper presents SimMTM, a simple pre-training framework for masked time series modeling. By relating masked modeling to manifold learning, SimMTM proposes to recover masked time points by the weighted aggregation of multiple neighbors outside the manifold, which eases the reconstruction task by assembling ruined but complementary temporal variations from multiple masked series. **SimMTM achieves state-of-the-art fine-tuning performance compared to 6 advanced baselines in 12 well-established benchmarks in two canonical time series analysis tasks: forecasting and classification, covering both in- and cross-domain settings.**

The reviewers generally held positive opinions of our paper, in that the proposed method is "**novel**", "**a novel representation learning method**", "**sound and well-motivated**", this paper has "**comprehensive comparative experiments**", and it archives "**remarkable performance improvements**", and the empirical evaluation is "**comprehensive and convincing**".

The reviewers also raised insightful and constructive concerns. We have made every effort to address all the concerns by providing sufficient evidence and requested results. Here is the summary of the major revisions:

- **Clarify the unified backbone setting (Reviewer xbHA)**: For all baselines and all tasks, we have conducted two types of experiments in the original submission, including unified backbone and the specific backbone proposed in their original papers. Furthermore, we perform experiments by keeping each part of the dual-tower TF-C consistent to SimMTM. All results show SimMTM produce the best performance.

- **Resolve the writing issues (Reviewer ZNHw, L1Yx, 5Qee)**: We have rewritten or modified some formulas and corrected some writing errors to improve presentation quality.

- **Analyze the relationship of masked ratios and numbers (Reviewer ZNHw)**: We've proven that one masked series with a lower masked ratio can perform well, and we have further analyzed the relationship between masked ratio and masked numbers. Results demonstrate that the design of multiple masked series aggregation in SimMTM is critical, and a reasonable balance between masked ratio and masked numbers shows better performance.

- **Add comparison in different masking rules (Reviewer ZNHw)**: We further explored the effect of different masking rules, including fixed position (tail or head) and random position (large, small segment or points) masking. Results show random masking performs better than fixed masking, and the difference is insignificant among different random masking.

- **Analysis the effects of different reconstruction candidates (Reviewer 5Qee)**: We have conducted a comparison experiment by aggregating from own or all masked series in the reconstruction process. Results show using both own and other less-related noise series for reconstruction performs better.

- **Add baselines (Reviewer ZNHw)**: By comparing SimMTM and three generalized representation learning methods TCL(2016), CPC(2018), and PCL(2019) in forecasting and classification tasks, we further demonstrate that SimMTM achieves state-of-the-art performance.

The valuable suggestions from reviewers are very helpful for us to improve our paper. We'd be very happy to answer any further questions.

Looking forward to the reviewer's feedback.

#### **The mentioned Tables and Figures are included in the following PDE file.**

- **Figure 1**: More showcases for Reviewer xbHA.
- **Tabel 1**: New baselines in forecasting for Reviewer ZNHw.
- **Tabel 2**: New baselines in classification for Reviewer ZNHw.
- **Tabel 3**: Different masking rules for Reviewer ZNHw.

---

### Comment · Area_Chair_mxUn · 2023-08-21
**Mixed reviews**

Dear reviewers (and especially reviewer ZNHw),

Thanks for the hard work so far!

This paper received mixed reviews and the authors have responded multiple times.

We need to ideally reach a consensus in the rebuttal period, and at least should have active discussions, updated recommendations, and acknowledgment that you have read the response.

Can you please check other reviews and the author rebuttal and see if your opinion has changed? Please give your reasoning in as much detail as possible.

AC

---

### Decision · Program_Chairs · 2023-09-21

**Decision:**

Accept (spotlight)

**Comment:**

Post rebuttal, all of reviewers lean toward acceptance. The AC checked all the materials and concurs that the paper has done a valuable exploration of the "masked-modeling" idea in time-series data, addressing issues encountered by the simple baseline and providing sufficient experimental validations. Thus the work should be accepted. Please incorporate necessary changes in the final version.